# UNCERTAINTY-AWARE SCHEDULING: STATE-DEPENDENT TRAINING-FREE DIFFUSION ALIGNMENT

## ABSTRACT

Training-free guidance has emerged as a promising approach to aligning diffusion models with downstream objectives by steering the denoising process with reward-based signals. Typically, reward functions are trained on clean images then applied to noisy intermediate predictions ($\hat{x}_{0|t}$), suffering from a domain gap that compromises effective guidance. Existing methods improve guidance through handcrafted schedules, such as fixed or time-dependent tempering. However, such state-agnostic scheduling is prone to suboptimal alignment as we discovered that denoising progress varies widely across samples. We propose State-Dependent Adaptive Guidance (SDAG) to schedule guidance through an uncertainty-aware confidence-calibrated assessment. SDAG introduces a lightweight quality predictor that estimates denoising progress from intermediate states, i.e. the closeness between their approximated and the final clean targets. Through a last-layer Laplace approximator, this predictor provides uncertainty estimates, which are used together with the closeness scores to scale guidance reliably. Our SDAG applies to both standard denoising and population-based sampling, such as Sequential Monte Carlo, where coordination by effective sample size ensures robust collective guidance. Experiments demonstrate that SDAG achieves superior alignment while maintaining computational efficiency, establishing a promising paradigm for adaptive guidance in training-free diffusion alignment.

## 1 INTRODUCTION

Diffusion models have become a standard backbone of modern image generation, achieving state-of-the-art fidelity and controllability in text-to-image synthesis (Ho et al., 2020; Song et al., 2020b; Saharia et al., 2022; Rombach et al., 2022). Yet aligning these models to diverse user preferences and downstream objectives remains a core challenge. Recently, training-free test-time guidance has shown promise in enabling generation to be steered by external objectives without model retraining nor prohibitive overhead (Poole et al., 2022; Ye et al., 2024).

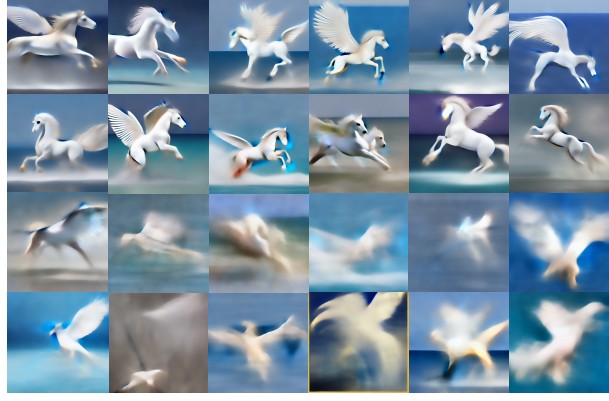

Figure 1: Denoising heterogeneity at fixed timestep $t(=781)$. Predicted clean images $\hat{x}_{0|t}$ (Eq. (3)) exhibit significant variance in denoising process across different samples.

Current training-free guidance (TFG) methods typically align diffusion models by guiding an intermediate denoising of state $x_t$ through a pre-trained reward function $R(\cdot)$, which is defined on the image space (Ye et al., 2024). A common practice (Chung et al., 2022; Song et al., 2023; He et al., 2023) is to first approximate a clean target $\hat{x}_{0|t}$ given noisy $x_t$ using the Tweedie's formula (Stein, 1981; Efron, 2011). Then, an approximated clean target is evaluated by the reward function so as to steer the denoising of the noisy state: $x_{t-1} =$

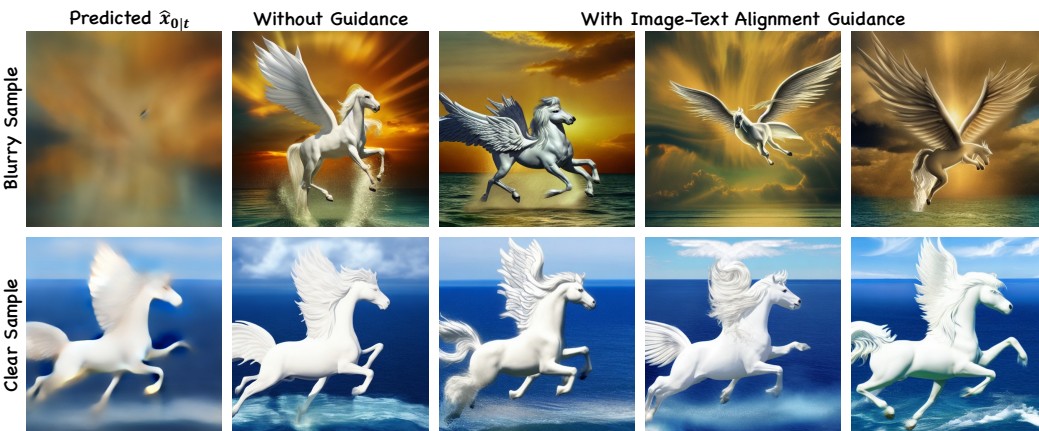

Figure 2: Fixed guidance strength exhibits differential effectiveness based on denoising progress. Under identical guidance strength, blurry samples (top row) show substantial improvement from guidance (columns 3-5 vs. column 2), while clear samples (bottom row) show minimal response. This reveals that fixed guidance schedules are suboptimal: guidance should be applied more aggressively to clear samples that can benefit from stronger corrections.

$\epsilon_\theta(\boldsymbol{x}_t, t) + \lambda_t \nabla_{\boldsymbol{x}_t} \log R(\hat{\boldsymbol{x}}_{0|t})$, where $\epsilon_\theta(\cdot)$ is the diffusion denoiser and $\lambda_t$ controls the strength of the reward guidance.

However, the reward function $R(\cdot)$ is typically trained on clean images then applied to intermediate noisy predictions, resulting in a domain gap. This gap causes unreliable reward evaluation of noisy predictions, making the guidance steering difficult. To compensate for this unreliability, prior works resort to handcrafted or heuristic schedules, e.g., (Ye et al., 2024; Kim et al., 2025) fixing guidance strength globally, scaling it only with the timestep, or tempering it in pre-defined patterns.

Despite these advances, such state-agnostic schedules ignore the fact that denoising progress varies widely across samples, even at the same timestep. More specifically, as illustrated in Fig. 1, samples at a given timestep exhibit significant variance in their denoising progress; some may already display clear structure with high reward potential, while others remain largely incoherent. This heterogeneity reveals that diverse states require adaptive guidance schedules rather than a single, shared schedule currently in place. As a result, guidance is often misapplied: samples that are already coherent receive unnecessary refinement, while more degraded states fail to obtain sufficient support. Thus, the fixed or time-dependent scheduling ultimately leads to ineffective guidance.

To address this, we propose State-Dependent Adaptive Guidance (SDAG), a training-free alignment that provides reliable, adaptive guidance scheduling for heterogeneous denoising processes. Our key insight is that scheduling should be coupled with the reliability of quality estimates: guidance is intensified only when intermediate states are trustworthy and promising, while unreliable states are treated conservatively, overcoming the drawback of heuristic, state-agnostic schedules. We introduce a surrogate module, a lightweight quality predictor, which predicts the similarity, i.e. closeness, between the approximated ($\hat{\boldsymbol{x}}_{0|t}$) and the final clean outputs ($\boldsymbol{x}_0$). This closeness is used as an indicator to scale the guidance strength for each state adaptively. To further enhance the reliability of this indicator, we equip the predictor with a last-layer Laplace approximator (Daxberger et al., 2021) that estimates the predictive uncertainty. We transform these uncertainty estimates into Lower Confidence Bounds (LCB) (Nocedal & Wright, 2006; Brochu et al., 2010), which provide confidence-calibrated assessments for adaptive line search (Armijo, 1966; Nocedal & Wright, 2006), thereby scaling strength coefficients reliably (Abdar et al., 2021).

We summarize our contributions as follows:

- We present State-Dependent Adaptive Guidance (SDAG), which provides adaptive, reliable guidance scheduling with uncertainty quantification in training-free diffusion alignment.
- SDAG applies to both the standard denoising and population-based sampling, such as Sequential Monte Carlo (Doucet et al., 2001), where effective sample size ensures robust collective guidance.
- Comprehensive experiments demonstrate superior alignment performance across diverse reward functions while maintaining computational efficiency.

## 2 RELATED WORK

**Training-free Guidance and Adaptive Strategies.** Training-free guidance enables conditioning pre-trained diffusion models (Rombach et al., 2022; Esser et al., 2024; Wan et al., 2025; Kong et al., 2020; Liu et al., 2023) on target properties using external predictors without model retraining Ho et al. (2020); Song et al. (2020b). Early approaches include DPS (Chung et al., 2022), LGD (Song et al., 2023), and MPGD (He et al., 2023). Universal Guidance (Bansal et al., 2023) provides a unified framework for backward optimization, while TFG (Ye et al., 2024) systematically benchmarks existing approaches as special cases within a common design space. Recently, several works have recognized that constant guidance weights are suboptimal, leading to adaptive guidance strategies. Muse (Chang et al., 2023) and MDTv2 (Gao et al., 2023) proposed increasing guidance coefficients at higher noise levels, while Kynkäänniemi et al. (2024) demonstrated that guidance is harmful at high noise levels, unnecessary at low noise levels, and only beneficial in the middle range. Semantic-aware classifier-free guidance (Shen et al., 2024) introduced spatially varying guidance weights to address artifacts. More principled frameworks include adaptive diffusion guidance (Azangulov et al., 2025), which formulated guidance scheduling as a stochastic optimal control problem, deriving time- and state-dependent policies. However, these adaptive approaches either adjust conditional model weights within CFG framework rather than external reward guidance strength, or show only marginal improvements due to idealized assumptions.

**Sequential Monte Carlo (SMC) for Diffusion** has gained attention as a principled approach to guidance and alignment. The Feynman-Kac framework (Skreta et al., 2025) developed SMC correctors to ensure unbiased sampling from guided distributions, providing theoretical grounding for SMC-based guidance. Diffusion Alignment by Sampling (DAS) (Kim et al., 2025) achieved test-time alignment with arbitrary reward functions while preserving diversity and avoiding reward over-optimization. Recent work has expanded SMC applications across different domains and settings. Monte Carlo guided denoising (Cardoso et al., 2023) applied SMC to Bayesian linear inverse problems, while particle filtering frameworks (Liu et al., 2024) corrected distributional discrepancies in diffusion generation. Particle guidance (Corso et al., 2023) introduced diverse sampling methods, and debiasing guidance (Lee et al., 2025) extended SMC to discrete diffusion models. Particle Gibbs sampling (Dang et al., 2025) explored inference-time scaling in discrete diffusion language models, while Yoon et al. (2025) presented posterior-aware particle initialization. In parallel to SMC-based methods, CoDe (Singh et al., 2025) proposes a blockwise control strategy at intermediate denoising steps to align sampling with downstream rewards in text-to-image tasks. SVDD (Li et al., 2024) introduces derivative-free value-based decoding to optimize downstream reward functions in both continuous and discrete diffusion models across images, molecules, and biological sequences. Despite these advances, existing SMC approaches rely on fixed tempering schedules that cannot adapt to the evolving quality and uncertainty of particle populations during generation.

## 3 BACKGROUND

**Diffusion model preliminaries.** Let $p_\theta(\boldsymbol{x}_0 \mid c)$ be a pre-trained diffusion model that generates data $\boldsymbol{x}_0$ (e.g., an image) conditioned on some context $c$ (e.g., a text prompt), where $\theta$ parameterizes the diffusion model. The model (Ho et al., 2020; Song et al., 2020b) is built upon a forward diffusion process that gradually corrupts data by adding Gaussian noise according to a predefined variance schedule $\{\beta_t\}_{t=1}^T$ where $\beta_t \in (0, 1)$:

$$q(\boldsymbol{x}_t \mid \boldsymbol{x}_{t-1}) = \mathcal{N}(\boldsymbol{x}_t; \sqrt{1 - \beta_t}\boldsymbol{x}_{t-1}, \beta_t I) \tag{1}$$

The neural network $\epsilon_\theta$ is trained to predict the noise added at each step, enabling the reverse process:

$$p_\theta(\boldsymbol{x}_{t-1} \mid \boldsymbol{x}_t, c) = \mathcal{N}\left(\boldsymbol{x}_{t-1}; \frac{1}{\sqrt{\alpha_t}}\left(\boldsymbol{x}_t - \frac{\beta_t}{\sqrt{1 - \bar{\alpha}_t}}\epsilon_\theta(\boldsymbol{x}_t, c, t)\right), \sigma_t^2 I\right) \tag{2}$$

where $\alpha_t = 1 - \beta_t$, $\bar{\alpha}_t = \prod_{s=1}^t \alpha_s$ and $\sigma_t^2 = \tilde{\beta}_t = \frac{1-\bar{\alpha}_{t-1}}{1-\bar{\alpha}_t}\beta_t$. Thanks to Tweedie's formula (Stein, 1981; Efron, 2011), we yield an estimate of the clean data from any noisy intermediate state using the noise prediction:

$$\hat{\boldsymbol{x}}_{0|t} = \frac{\boldsymbol{x}_t - \sqrt{1 - \bar{\alpha}_t}\epsilon_\theta(\boldsymbol{x}_t, c, t)}{\sqrt{\bar{\alpha}_t}} \tag{3}$$

which serves as the foundation for applying reward functions during the generation process.

**Sequential Monte Carlo (SMC) for guided diffusion** reformulates training-free guidance (Ye et al., 2024) as tempered importance sampling over a population of diffusion trajectories. Rather than evolving a single sample, SMC maintains $N$ particles that are iteratively denoised using the pretrained model while gradually biasing the sampling distribution toward a reward-weighted target distribution $p_\theta(\boldsymbol{x}_0|c)\exp\{\gamma R(\boldsymbol{x}_0)\}$, where $R(\cdot)$ is a reward function and $\gamma := \frac{\lambda}{\alpha}$ controls the strength of reward bias (Doucet et al., 2001). At each timestep $t$, the algorithm proposes transitions $\boldsymbol{x}_{t-1}^i \sim p_\theta(\boldsymbol{x}_{t-1}|\boldsymbol{x}_t^i, c)$ for each particle and assigns incremental importance weights:

$$w_{t-1}^i \propto \exp\left\{\gamma_t R\left(\hat{\boldsymbol{x}}_{0|t}^i\right)\right\} \tag{4}$$

where $\hat{\boldsymbol{x}}_{0|t}^i$ is derived using Eq. 3. These weights are normalized and particles are resampled when the effective sample size falls below ESS constraint (Liu & Chen, 1998), thereby maintaining a balance between exploration of diverse hypotheses and exploitation of high-reward regions.

**Problem Definition** Given a pre-trained diffusion model $p_\theta(\boldsymbol{x}_0|c)$ and an external reward function $R : \mathcal{X} \to \mathbb{R}$ that evaluates the quality or desirability of generated samples, the goal is to sample from a modified distribution that favors high-scoring outputs under $R(\cdot)$ while preserving the generative quality of the original model. Formally, we aim to sample from a target distribution (Ho et al., 2020):

$$q^*(\boldsymbol{x}_0|c) \propto p_\theta(\boldsymbol{x}_0|c)\exp\{\lambda R(\boldsymbol{x}_0)\} \tag{5}$$

where $\lambda > 0$ controls the strength of the bias toward high-reward samples. Since direct sampling from this distribution is intractable, training-free guidance methods approximate this by modifying the reverse diffusion process. Specifically, the complete denoising update can be decomposed into three complementary components:

$$\boldsymbol{x}_{t-1} = \underbrace{\beta_t \boldsymbol{x}_t - \gamma_t \epsilon_\theta(\boldsymbol{x}_t, \varnothing) + \sigma_t \epsilon}_{\text{Unconditional Denoising}} + \underbrace{\gamma_t\Big(w\left[\epsilon_\theta(\boldsymbol{x}_t, c) - \epsilon_\theta(\boldsymbol{x}_t, \varnothing)\right]\Big)}_{\text{Classifier-free Guidance}} + \underbrace{\lambda_t \sigma_t \nabla_{\boldsymbol{x}_t} \log R(\hat{\boldsymbol{x}}_{0|t})}_{\text{Reward-based Guidance}} \tag{6}$$

where unconditional denoising provides the foundational sampling dynamics, classifier-free guidance (Ho & Salimans, 2022) incorporates conditioning information $c$ with guidance scale $w$, and reward-based corrections steer generation toward high-reward regions with guidance strength $\lambda_t$. The central challenge lies in determining appropriate values of $\lambda_t$ that ensure harmonious interaction with both the unconditional diffusion process and existing classifier-free guidance mechanism, thereby preserving sample quality and diversity while achieving effective reward optimization.

## 4 STATE-DEPENDENT ADAPTIVE GUIDANCE

This section introduces State-Dependent Adaptive Guidance (SDAG), a training-free alignment method that dynamically determines guidance strength $\lambda_t$ through confidence-aware individual optimization and population-level coordination within the Sequential Monte Carlo framework.

### 4.1 CONFIDENCE-AWARE INDIVIDUAL OPTIMIZATION

**Quality Predictor** To enable general applicability across diverse reward functions, we design a quality predictor $Q(\boldsymbol{x}_t, t)$ that assesses denoising progress rather than task-specific rewards. During training data collection, we save diffusion trajectories and compute denoising quality labels as:

$$y_t = \frac{\phi(\hat{\boldsymbol{x}}_{0|t}) \cdot \phi(\boldsymbol{x}_0)}{|\phi(\hat{\boldsymbol{x}}_{0|t})||\phi(\boldsymbol{x}_0)|} \tag{7}$$

where $\phi(\cdot)$ denotes a pre-trained vision encoder, $\hat{\boldsymbol{x}}_{0|t}$ is the predicted clean image at timestep $t$ (Eq. 3), and $\boldsymbol{x}_0$ is the final generated sample. This cosine similarity between semantic embeddings serves as a universal measure of how well the intermediate prediction aligns with the final output's content, independent of any specific reward function. The predictor is trained to map intermediate diffusion states $(\boldsymbol{x}_t, t)$ to quality scores directly: $Q : (\boldsymbol{x}_t, t) \mapsto y_t$, assessing denoising progress from the latent representation without requiring decoder evaluation during inference. This state-conditioned assessment enables differentiated guidance based on denoising progress, where high-similarity states exhibit clear structure while low-similarity states remain blurry. More visual examples and qualitative analysis refer to Fig. 8 and Sec. D.3.

However, the quality predictor output represents semantic similarity rather than guidance strength, requiring a robust mechanism to translate quality estimation into appropriate step sizes $\lambda_t$. To bridge this gap, we equip the quality predictor with uncertainty quantification through Last-Layer Laplace approximation (Mackay, 1992; Daxberger et al., 2021), which efficiently provides epistemic uncertainty estimates $\sigma(\boldsymbol{x}_t, t)$ by approximating the posterior over the last layer parameters using a Laplace approximation around the MAP estimate. We combine mean predictions with uncertainty estimates to form Lower Confidence Bounds (LCB) that provide conservative quality assessments:

$$\text{LCB}(\boldsymbol{x}_t, t) = Q(\boldsymbol{x}_t, t) - \kappa \sigma(\boldsymbol{x}_t, t) \tag{8}$$

where $\kappa > 0$ controls the conservativeness of the bound. We then employ this quality assessment in a systematic line search procedure (Armijo, 1966; Nocedal & Wright, 2006) to determine appropriate guidance strength for each particle.

**Definition 4.1.** (Per-particle Confidence-Aware Line Search) *Given reward gradient direction $\hat{v} = \nabla_{\boldsymbol{x}_t} \log R(\hat{x}_{0|t})$, we determine the maximum safe step $\lambda^{\max}$ using a confidence-aware Armijo backtracking rule. Specifically, starting from a given initial step size $\lambda^{init}$ at timestep t, we find the largest $\lambda_t$ satisfying:*

$$\text{LCB}(\boldsymbol{x}_t + \lambda_t \hat{v}) \geq \text{LCB}(\boldsymbol{x}_t) + c_1 \lambda_t \nabla_{\boldsymbol{x}_t} Q(\boldsymbol{x}_t, t)^T \hat{v} \tag{9}$$

*where $c_1$ is the Armijo constant (Armijo, 1966) and the gradient $\nabla_{\boldsymbol{x}_t} Q$ is computed via automatic differentiation through the quality predictor.*

Definition 4.1 ensures monotonic progress in quality assessment while maximizing improvement along the reward gradient direction. Particles in high-confidence regions (low $\sigma$) can take larger steps, while those in uncertain regions receive a more conservative treatment. The Armijo condition provides additional safeguards against overly aggressive steps even in confident regions, ensuring robust convergence properties.

**Adaptive Upper Bound Determination ($\boldsymbol{\lambda^{init}}$)** We establish the per timestep initial step size $\lambda^{init}$ by leveraging the magnitude of classifier-free guidance corrections as a natural reference scale. Given the decomposed denoising update formula Eq. 6, the CFG guidance term provides a heuristic upper bound:

$$\lambda^{init}(\boldsymbol{x}_t, c, t) = \frac{\|\gamma_t[w(\epsilon_\theta(\boldsymbol{x}_t, c) - \epsilon_\theta(\boldsymbol{x}_t, \varnothing))]\|_2}{\|\sigma_t \nabla_{\boldsymbol{x}_t} \log R(\hat{x}_{0|t})\|_2} \tag{10}$$

where $\|\cdot\|_2$ denotes the L2 norm. This adaptive initialization captures the model's learned understanding of appropriate correction magnitudes for each intermediate state, providing a principled starting point for the line search while preventing excessive reward guidance that could destabilize the generation process.

**Robustness to Predictor Errors** A key practical concern is how quality prediction errors affect guidance stability, particularly since ground truth labels are unavailable during inference. Our framework provides inherent robustness through the Armijo sufficient decrease condition.

Recent theoretical work shows that Armijo line search methods maintain convergence guarantees even under noisy objective evaluations (Vaswani & Babanezhad, 2025). This robustness property directly applies to our setting: when $Q(\boldsymbol{x}_t, t)$ overestimates denoising quality, the Armijo condition in Definition 4.1 automatically rejects overly aggressive steps that fail to achieve expected LCB improvement. Conversely, when quality is underestimated, the adaptive upper bound $\lambda^{init}$ from Eq. 10 ensures that line search starts from a reasonable scale derived from classifier-free guidance (Ho & Salimans, 2022; Yu et al., 2023) guidance magnitude, leading to conservative but safe step sizes that maintain sampling stability. Importantly, this mechanism operates entirely at inference time with minimal additional overhead, as the Last-Layer Laplace approximation (Daxberger et al., 2021) requires only a single forward pass through the pre-trained predictor.

Our empirical analysis reveals that this theoretical robustness manifests as oscillatory guidance patterns, where aggressive optimization phases are naturally followed by conservative recovery periods (Fig. 3, last column). The correction-recovery cycle demonstrates the ability of the model to self-regulate under predictor uncertainty, enabling effective reward optimization while maintaining sampling stability. More quality predictor details and prediction accuracy can be found in Sec. C.

---

**Algorithm 1** State-Dependent Adaptive Guidance (SDAG)

---

**Require:** Pre-trained diffusion model $\epsilon_\theta$, quality predictor $Q(\cdot)$ with uncertainty head approximated by Last Layer Laplace (LLL), reward function $R(\cdot)$
**Require:** Particles $N$, timesteps $T$, line search factor $s$, text prompt $c$
1: Initialize particles $\{\boldsymbol{x}_t^i\}_{i=1}^N \sim \mathcal{N}(0, I)$
2: **for** $t = T, T-1, \ldots, 1$ **do**
3:      **for** $i = 1, \ldots, N$ **do**             // Individual particle optimization (in parallel)
4:          $\lambda_i^{\max} \leftarrow$ Largest $\lambda = s^k \lambda_i^{init}$, $k = 0, ..., 10$ satisfying Armijo condition $\triangleright$ Definition 4.1
5:      **end for**
6:      **for** $i = 1, \ldots, N$ **do**                         // Population coordination
7:          $\boldsymbol{x}_{t-1}^i \leftarrow$ Apply Eq. (6) with guidance $\lambda_i^{\text{actual}} \leftarrow \min(\lambda_i^{\max}, \lambda^*)$      $\triangleright$ Definition 4.2
8:          $w_i \leftarrow \exp\{\gamma_t R(\hat{x}_{0|t-1}^i)\}$                      $\triangleright$ SMC importance weight
9:      **end for**
10:     If ESS $< \delta N$: Resample particles $\{\boldsymbol{x}_{t-1}^i\}$ according to normalized weights $\{w_i\}$
11: **end for**
12: **return** Final samples $\{\boldsymbol{x}_0^i\}_{i=1}^N$

---

## 4.2 Population-Level Coordination

The challenge of determining optimal guidance parameters extends beyond individual particle optimization to population-level coordination. While each particle determines its own maximum safe step $\lambda_{i,t}^{\max}$ based on local uncertainty at timestep $t$, the resulting distribution $\{\lambda_{i,t}^{\max}\}_{i=1}^N$ exhibits significant heterogeneity that can destabilize sampling if left uncoordinated. The SMC framework provides a natural solution through population-level constraints that synthesize individual insights while maintaining theoretical convergence guarantees:

**Lemma 4.2.** (SMC Finite Sample Effectiveness) *For SMC with adaptive resampling controlling effective sample size ESS $\geq \delta N$, the algorithm achieves $O(1/\sqrt{N})$ convergence to the target distribution without requiring bounded importance weights, where convergence depends on local rather than global mixing properties (Huggins & Roy, 2015; Lee & Santana-Gijzen, 2024).*

This convergence guarantee enables effective population coordination even with modest particle counts: unlike traditional methods, e.g., MCMC algorithms requiring global spectral gap bounds (Levin & Peres, 2017) and parallel/simulated tempering requiring global mixing across temperature levels (Woodard et al., 2009), the local convergence properties of SMC allow us to design adaptive population-level constraints that synthesize individual insights into stable collective decisions. However, when only a small set of particles is used, SMC suffers from mode collapse (Hinne, 2025) and biased sampling. To address this, we introduce a guidance schedule that prevents mode collapse while ensuring stable sampling progress.

**Definition 4.2.** (Population Constraints) *Given individual safe steps $\{\lambda_i^{\max}\}_{i=1}^N$, the global coordination parameter is determined through dual constraints:*

$$\lambda^* = \min\left(\boldsymbol{\lambda_{\text{ESS}}} := \arg\max_\lambda\{\lambda : \text{ESS}(\lambda) \geq \delta N\}, \boldsymbol{\lambda_{\text{quant}}} := \text{Quantile}_\rho\left(\{\lambda_i^{\max}\}_{i=1}^N\right)\right) \quad (11)$$

The ESS constraint ($\boldsymbol{\lambda_{\text{ESS}}}$) ensures that at least a fraction $\delta$ of particles remain effective after reweighting to prevent mode collapse and particle impoverishment. The quantile constraint ($\boldsymbol{\lambda_{\text{quant}}}$) ensures that a proportion $\rho$ of particles can feasibly take the chosen global step. This dual-constraint optimization naturally adapts to population heterogeneity: when particles exhibit similar assessments, both constraints align; when the population is highly heterogeneous, the algorithm automatically becomes more conservative to maintain collective stability. Each particle's actual step becomes $\lambda_i^{\text{actual}} = \min(\lambda_i^{\max}, \lambda^*)$, balancing individual safety with collective stability. The pseudo-code of the entire algorithm is shown in Algo. 1.

## 5 Experiments

**Reward Functions for Guidance.** We conduct experiments using four diverse reward functions as guidance objectives: (1) **Aesthetic Score** (Schuhmann, 2022), which evaluates visual appeal and

Table 1: Comparative evaluations of state-dependent adaptive guidance across multiple reward functions and conditioning strategies. We compared our SDAG method against three baseline approaches (FreeDoM, MPGD, DAS) using diverse reward metrics including Aesthetic, CLIPScore, HPSv2, PickScore, ImageReward, TCE (Truncated CLIP Entropy), and MPD (Mean Pairwise Distance). These evaluations have been organized by presenting target reward functions first followed by complementary metrics. **Bold** indicates best performance, **underlined bold** indicates second best.

| Methods | FreeDoM (Yu et al., 2023) | | MPGD (He et al., 2023) | | DAS (Kim et al., 2025) | | Ours | |
|---|---|---|---|---|---|---|---|---|
| | Aesthetic | CLIPScore | Aesthetic | CLIPScore | Aesthetic | CLIPScore | Aesthetic | CLIPScore |
| **Aesthetic** (↑) | $5.9319_{\pm0.40}$ | $5.2799_{\pm0.33}$ | $5.5065_{\pm0.35}$ | $\mathbf{5.2950_{\pm0.36}}$ | $\underline{\mathbf{6.0375}}_{\pm0.35}$ | $5.2279_{\pm0.35}$ | $\mathbf{6.1860_{\pm0.49}}$ | $\underline{\mathbf{5.2815}}_{\pm0.34}$ |
| **CLIPScore** (↑) | $0.2577_{\pm0.04}$ | $\mathbf{0.3029_{\pm0.03}}$ | $0.2574_{\pm0.03}$ | $0.2628_{\pm0.03}$ | $\underline{\mathbf{0.2596}}_{\pm0.04}$ | $0.2782_{\pm0.03}$ | $0.2645_{\pm0.03}$ | $\underline{\mathbf{0.2797}}_{\pm0.03}$ |
| HPSv2 (↑) | $0.2782_{\pm0.02}$ | $0.2795_{\pm0.02}$ | $\underline{\mathbf{0.2783}}_{\pm0.02}$ | $0.2782_{\pm0.01}$ | $0.2777_{\pm0.02}$ | $\mathbf{0.2803_{\pm0.02}}$ | $\mathbf{0.2800_{\pm0.01}}$ | $\underline{\mathbf{0.2798}}_{\pm0.01}$ |
| PickScore (↑) | $\underline{\mathbf{0.2187}}_{\pm0.01}$ | $0.2190_{\pm0.01}$ | $0.2186_{\pm0.01}$ | $0.2195_{\pm0.01}$ | $0.2184_{\pm0.01}$ | $\mathbf{0.2203_{\pm0.01}}$ | $\mathbf{0.2213_{\pm0.01}}$ | $\underline{\mathbf{0.2201}}_{\pm0.01}$ |
| ImageReward (↑) | $0.2183_{\pm1.16}$ | $0.5023_{\pm1.04}$ | $\underline{\mathbf{0.4542}}_{\pm0.97}$ | $0.4058_{\pm1.00}$ | $0.3342_{\pm1.17}$ | $\mathbf{0.5976_{\pm1.08}}$ | $\mathbf{0.5989_{\pm1.00}}$ | $\underline{\mathbf{0.5478}}_{\pm1.02}$ |
| TCE (↑) | 1.4832 | **1.4917** | 1.4750 | $\underline{\mathbf{1.4709}}$ | $\underline{\mathbf{1.4932}}$ | 1.4698 | **1.5112** | 1.4569 |
| MPD (↑) | 0.8307 | 0.8205 | 0.8188 | 0.8168 | $\underline{\mathbf{0.8494}}$ | $\underline{\mathbf{0.8368}}$ | **0.8558** | **0.8512** |

(a) Performance comparison under **Aesthetic** and **CLIPScore** guidance. Our method achieves superior reward alignment while maintaining competitive performance on complementary metrics, demonstrating effective guidance adaptation without sacrificing generation quality.

| Methods | FreeDoM (Yu et al., 2023) | | MPGD (He et al., 2023) | | DAS (Kim et al., 2025) | | Ours | |
|---|---|---|---|---|---|---|---|---|
| | HPSv2 | PickScore | HPSv2 | PickScore | HPSv2 | PickScore | HPSv2 | PickScore |
| **HPSv2** (↑) | $\mathbf{0.2891_{\pm0.01}}$ | $\mathbf{0.2833_{\pm0.02}}$ | $0.2778_{\pm0.01}$ | $0.2779_{\pm0.01}$ | $0.2842_{\pm0.01}$ | $0.2823_{\pm0.01}$ | $\underline{\mathbf{0.2845}}_{\pm0.01}$ | $\underline{\mathbf{0.2828}}_{\pm0.01}$ |
| **PickScore** (↑) | $\mathbf{0.2234_{\pm0.01}}$ | $\mathbf{0.2286_{\pm0.01}}$ | $0.2192_{\pm0.01}$ | $0.2193_{\pm0.01}$ | $0.2219_{\pm0.01}$ | $0.2234_{\pm0.01}$ | $\underline{\mathbf{0.2228}}_{\pm0.01}$ | $\underline{\mathbf{0.2239}}_{\pm0.01}$ |
| Aesthetic (↑) | $\mathbf{5.3808_{\pm0.29}}$ | $5.2946_{\pm0.34}$ | $5.3094_{\pm0.36}$ | $\mathbf{5.3166_{\pm0.34}}$ | $5.2820_{\pm0.33}$ | $5.2426_{\pm0.31}$ | $\underline{\mathbf{5.3277}}_{\pm0.31}$ | $\underline{\mathbf{5.3105}}_{\pm0.26}$ |
| CLIPScore (↑) | $0.2602_{\pm0.03}$ | $0.2626_{\pm0.03}$ | $0.2627_{\pm0.03}$ | $0.2622_{\pm0.03}$ | $\underline{\mathbf{0.2686}}_{\pm0.03}$ | $0.2669_{\pm0.03}$ | $\mathbf{0.2690_{\pm0.03}}$ | $\mathbf{0.2705_{\pm0.03}}$ |
| ImageReward (↑) | $0.4524_{\pm1.16}$ | $0.3935_{\pm1.01}$ | $0.4249_{\pm0.96}$ | $0.4278_{\pm0.94}$ | $\mathbf{0.6047_{\pm1.07}}$ | $\underline{\mathbf{0.6753}}_{\pm1.00}$ | $\underline{\mathbf{0.6764}}_{\pm1.02}$ | $\mathbf{0.7326_{\pm1.01}}$ |
| TCE (↑) | 1.4549 | 1.4606 | 1.4684 | **1.4692** | 1.4781 | 1.4539 | $\underline{\mathbf{1.4697}}$ | $\underline{\mathbf{1.4610}}$ |
| MPD (↑) | 0.8233 | 0.8186 | 0.8166 | 0.8157 | $\underline{\mathbf{0.8483}}$ | **0.8420** | **0.8509** | $\underline{\mathbf{0.8329}}$ |

(b) Performance comparison under **HPSv2** and **PickScore** guidance. Results show consistent improvements in target reward optimization with maintained diversity metrics (TCE, MPD), validating the effectiveness of our adaptive guidance strategy across different reward functions.

image quality; (2) **CLIPScore** (Radford et al., 2021), measuring text-image alignment through contrastive language-image pre-training (Radford et al., 2021); (3) **HPSv2** (Wu et al., 2023), a human preference-based scoring model trained on human feedback data; and (4) **PickScore** (Kirstain et al., 2023), designed to predict human preferences for image generation tasks.

**Evaluation Protocol.** Following DAS (Kim et al., 2025), we measured performance on target reward functions and alternative human preference metrics ImageReward (Xu et al., 2023). We assessed generation diversity using Truncated CLIP Entropy (TCE) (Ibarrola & Grace, 2024), which quantifies CLIP embedding entropy, and Mean Pairwise Distance (MPD), which measures perceptual differences by LPIPS (Zhang et al., 2018). We compared against three training-free guidance methods that focus on guidance strength control: **FreeDoM** (Yu et al., 2023), employing energy-guided conditional generation; **MPGD** (He et al., 2023), implementing manifold-preserving guidance; and **DAS** (Kim et al., 2025), utilizing SMC for robust guidance optimization. These baselines represent the current leading approaches in adaptive guidance strength optimization for training-free diffusion model alignment.

**Implementation Details** We evaluated our method using stable-diffusion-v1-5 (Rombach et al., 2022; von Platen et al., 2022) as the backbone diffusion model. Our experimental design ensures rigorous comparison through controlled randomness: all methods used identical initial noise and fixed random generators for DDIM (Song et al., 2020a) sampling with $\eta = 1$, isolating the impact of guidance strength adaptation. Additional implementation details and hyperparameter sensitivity analysis are provided in Appendix C and D.5, respectively.

## 5.1 QUANTITATIVE ANALYSIS

Table 1 presents comprehensive evaluation results across diverse reward functions and evaluation metrics, demonstrating consistent superiority of our SDAG approach. We organized the evaluation metrics by presenting target reward functions first, followed by complementary metrics to highlight the trade-offs between reward optimization and quality preservation.

**Suboptimal Alignment Analysis** In Table 1a, where Aesthetic and CLIPScore serve as target rewards, we observe that FreeDoM achieves strong performance on target metrics (CLIPScore: **0.3029**) but exhibits degraded performance on complementary metrics, indicating over-optimization. Conversely, DAS demonstrates a contrasting characteristic, that is, a strong performance on non-

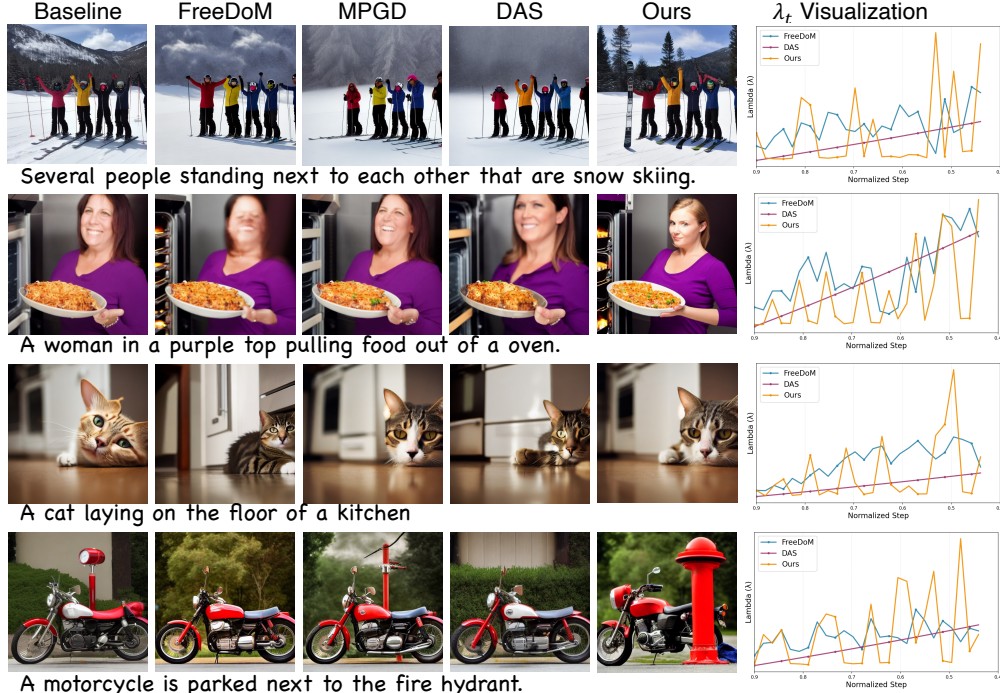

Figure 3: **Qualitative comparisons** across four text prompts demonstrating SDAG guidance effectiveness and $\lambda$ schedule patterns. Each row shows: baseline generation without guidance, followed by FreeDoM, MPGD, DAS, and SDAG (ours). Our approach achieves better text-image alignment (e.g., correct clothing colors, accurate object placement) with fewer visual artifacts. The **$\lambda$ Visualization** panels show SDAG's distinctive oscillatory guidance pattern that enables aggressive corrections followed by recovery phases, allowing more effective reward optimization compared to baseline methods' smoother, more conservative schedules.

target metrics (HPSv2: **0.2803**, ImageReward: **0.5976**) but weaker target reward optimization, suggesting under-optimization. These characteristics become more pronounced in Table 1b. FreeDoM achieves highest target scores but shows significant degradation in generation quality and diversity metrics. DAS maintains strong performance on complementary metrics but fails to fully exploit the target rewards. These consistent patterns in model performance characteristics demonstrate that existing methods struggle with the fundamental trade-off between reward optimization and quality preservation.

**Balanced Optimization.** SDAG consistently achieves superior performance against existing methods across both target and complementary metrics. In Table 1a, SDAG reaches the highest aesthetic scores while maintaining competitive CLIPScore alignment. Similarly, in Table 1b, SDAG achieves optimal HPSv2 and PickScore performance without sacrificing aesthetic quality or diversity. This balanced optimization proves that the proposed state-dependent adaptation addresses the fundamental limitation of fixed guidance schedules. Furthermore, selective temporal sampling enables SDAG to achieve these improvements with minimal additional computational cost, significantly reducing compute time without compromising model performance. (see Inference Time Analysis in Sec 5.3).

## 5.2 QUALITATIVE ANALYSIS

Fig. 3 provides visual evidence of SDAG's superior generation quality across multiple dimensions. Compared to baseline methods, our approach produces images with reduced artifacts (row 2 and 3), and enhanced visual diversity while maintaining semantic coherence with the input prompts (rows 2 and 4).

**Adaptive Guidance Pattern Analysis.** The $\lambda$ visualization in Fig. 3 reveals a distinctive oscillatory pattern in our method's guidance schedule, contrasting sharply with the monotonic trends of baseline approaches. This oscillation reflects a fundamental insight into the diffusion process: after applying reward-based corrections that perturb the current state $x_t$ away from its natural dis-

tribution, the denoising process requires recovery steps to realign with the learned manifold. Our adaptive schedule implements a "**correction-recovery**" cycle: aggressive guidance applications are followed by reduced intervention periods, allowing the model to integrate corrections while maintaining distributional coherence. This pattern enables more aggressive reward optimization during appropriate phases while preventing the accumulation of distributional drift that leads to artifacts in fixed-schedule methods. The oscillatory behavior demonstrates that effective guidance requires dynamic adaptation to the evolving denoising progress rather than monotonic strength progression.

## 5.3 COMPONENT ANALYSIS

**Number of Particles** Figure 4 demonstrates the relationship between particle count and generation quality across key metrics. As the number of particles increases from 1 to 16, we observe consistent improvements in both **CLIPScore** (text-image alignment) and Mean Pairwise Distance (**MPD**) (sample diversity). The gains are most pronounced between 1-4 particles, with diminishing returns beyond 8 particles. Based on this finding, we use 4 particles in our experiments to balance performance gains with computational efficiency. This validates that SMC's population-level

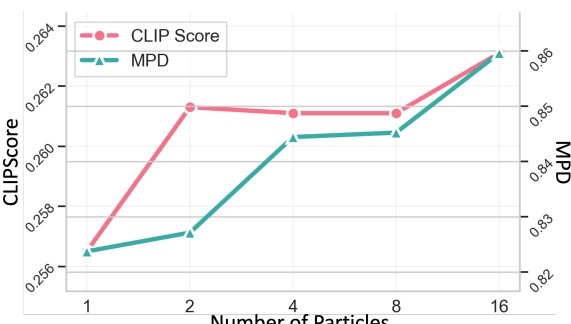

Figure 4: Image-Text alignment (CLIPScore) and diversity (MPD) scaling with number of particles.

coordination provides substantial benefits even with modest computational overhead (Lemma 4.2).

**Inference Time Analysis** While our method provides significant alignment improvements, SMC's computational overhead motivates selective temporal application. We explored applying **CLIPScore** guidance to different temporal ranges, where [1.0,0.0] stands for full coverage and [0.6,0.4] represents guid-

| Range | [1.0,0.0] | [0.9,0.1] | [0.8,0.2] | [0.7,0.3] | [0.6,0.4] |
|---|---|---|---|---|---|
| **CLIPScore** (↑) | 0.2703 | 0.2732 | 0.2738 | 0.2804 | 0.2811 |
| Aesthetic (↑) | 5.1909 | 5.2333 | 5.2260 | 5.2314 | 5.2315 |
| ImageReward (↑) | 0.4532 | 0.4905 | 0.5653 | 0.6834 | 0.7553 |
| MPD (↑) | 0.8394 | 0.8455 | 0.8374 | 0.8439 | 0.8449 |
| Runtime ($s$) | $34.78_{3.84\times}$ | $29.69_{3.27\times}$ | $23.91_{2.64\times}$ | $12.56^{*}_{1.38\times}$ | $12.50^{*}_{1.37\times}$ |

Table 2: Runtime analysis across temporal guidance ranges with performance validation. Runtime results are averaged over 50 runs with 4 particles. Baseline methods without SMC require 9.07s. $^{*}$ indicates performance saturation effects.

ance in a 20% intermediate window. All runtime measurements are averaged over 50 runs. Table 2 reveals a compelling efficiency-performance trade-off. Concentrating guidance to the narrow [0.6,0.4] window achieves dramatic computational speedup (2.8× compared to full application) while demonstrating improved CLIP alignment and substantially enhanced ImageReward scores (0.7553 vs. 0.4532). This counterintuitive improvement aligns with recent observations that applying guidance in limited temporal intervals can improve sample and distribution quality in diffusion models (Kynkäänniemi et al., 2024; Cao et al., 2025), suggesting that focused corrections during critical denoising phases are more effective than uniform temporal application. More in-depth analysis refer to Sec. D.3.

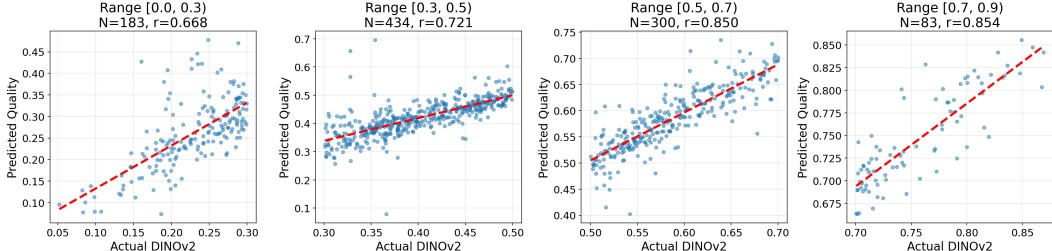

Figure 5: Stratified correlation analysis between predicted quality and actual DINOv2 similarity across different similarity ranges (N=1000, $t = 781$). The predictor maintains high correlation across all ranges, with particularly strong performance in medium-to-high similarity regions where quality discrimination is most critical.

**Quality Predictor Validation** Our adaptive guidance relies on accurate quality estimation at intermediate timesteps. To validate predictor performance, we analyze correlation between predicted quality and actual DINOv2 similarity on 1,000 samples at timestep $t = 781$. Fig. 6 shows strong predictive accuracy with overall Pearson $r = 0.955$ ($p < 0.001$), demonstrating that our lightweight predictor reliably estimates structural quality from noisy latent representations. Stratified analysis (Fig. 5) confirms consistent performance across similarity ranges: $r = 0.568$ for [0.0, 0.3], $r = 0.756$ for [0.3, 0.5], $r = 0.852$ for [0.5, 0.7], and $r = 0.821$ for [0.7, 0.9]. The predictor exhibits stronger performance in medium-to-high similarity regions where quality discrimination is most critical for adaptive guidance decisions. Expected Calibration Error (ECE = 0.015) indi-

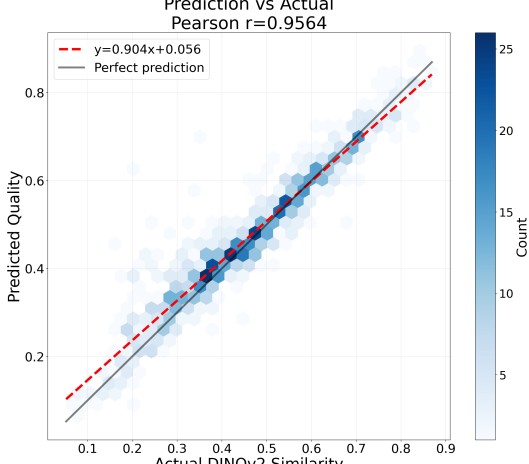

Figure 6: Overall correlation between predicted quality and actual DINOv2 similarity.

cates minimal systematic bias, ensuring reliable quality rankings for our LCB-based tempering strategy. For qualitative visualization of intermediate predictions $\hat{x}_{0|781}$ stratified by similarity scores, and uncertainty validation for LCB calibration, see Appendix D.3.

## 6 CONCLUSION

This work introduces a State-Dependent Adaptive Guidance (SDAG) method, which addresses guidance strength optimization through uncertainty quantification in training-free diffusion alignment, achieving best model performance with minimal additional computational cost. Our approach uses a lightweight quality predictor (8M) with epistemic uncertainty estimation to enable confidence-calibrated guidance decisions, extended to Sequential Monte Carlo for population-level coordination. Experiments demonstrate superior reward optimization while preserving generation quality, establishing adaptive guidance as a new paradigm for training-free diffusion alignment.

**Limitations & Future Work.** While our DINOv2-based quality predictor effectively captures structural coherence and denoising progress, the reward-agnostic design may not be sensitive to nuanced attributes that specific reward functions prioritize. Our approach assumes that structural coherence correlates with reward optimization potential, which holds for many common objectives but may be suboptimal for specialized reward functions focusing on domain-specific attributes, *e.g.* facial expression subtleties or fine-grained aesthetic preferences (Chan et al., 2022; Zhang et al., 2023b). Future work could explore task-adaptive quality predictors to address this limitation.

## REPRODUCIBILITY STATEMENT

To ensure that the proposed work is reproducible, we have included pseudocode for SDAG in Algo. 1. We have an explicit section (Sec. C) with implementation details, including visual encoder choice, hyperparameter settings, and training procedures.

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

# A  APPENDIX

This is the appendix for "Uncertainty-Aware Scheduling: State-Dependent Training-Free Diffusion Alignment". Tab. 3 summarizes the abbreviations and symbols used in the paper.

This appendix is organized as follows:

- Section B discusses the limitation and broader impact of our work.

- Section C presents additional details of our approach.

- Section D presents additional qualitative analysis.

Table 3: List of abbreviations and symbols used in the paper

| Symbol | Meaning |
|---|---|
| **Diffusion Model Symbols** | |
| $q(\cdot)$ | Distribution in the forward process |
| $p_\theta(\cdot)$ | Distribution in the $\theta$-parameterized reverse process |
| $\epsilon_\theta$ | Neural network for noise prediction |
| $c$ | Context/conditioning (e.g., text prompt) |
| $T$ | Total timesteps |
| $\beta_t, \{\beta_t\}_{t=1}^T$ | Variance schedule |
| $\alpha_t$ | $1 - \beta_t$ |
| $\bar{\alpha}_t$ | $\prod_{s=1}^t \alpha_s$ |
| $\sigma_t^2$ | Variance parameter |
| $\hat{\boldsymbol{x}}_{0\mid t}$ | Estimated clean data from noisy state at timestep $t$ |
| **Reward and Guidance Symbols** | |
| $R(\cdot)$ | Reward function |
| $\lambda, \lambda_t$ | Guidance strength parameter |
| $\lambda^*$ | Global coordination parameter |
| $\lambda^{init}$ | Initial step size |
| $\lambda_i^{\max}$ | Maximum safe step for particle $i$ |
| $\lambda_i^{\text{actual}}$ | Actual step size for particle $i$ |
| $\lambda_{\text{ESS}}$ | ESS-constrained guidance parameter |
| $\lambda_{\text{quant}}$ | Quantile-constrained guidance parameter |
| **SMC and Population Symbols** | |
| $N$ | Number of particles |
| $\gamma, \gamma_t$ | Reward bias strength $\frac{\lambda}{\alpha}$ |
| $w_t^i, w_i$ | Importance weight of particle $i$ |
| $\delta$ | ESS threshold fraction |
| $\rho$ | Quantile constraint proportion |
| **Quality Predictor Symbols** | |
| $Q(\boldsymbol{x}_t, t)$ | Quality predictor function |
| $y_t$ | Quality label (cosine similarity) |
| $\phi(\cdot)$ | Pre-trained vision encoder |
| $\sigma(\boldsymbol{x}_t, t)$ | Uncertainty estimate |
| $\kappa$ | Conservativeness parameter for LCB |
| $\hat{v}$ | Reward gradient direction $\nabla_{\boldsymbol{x}_t} \log R(\hat{x}_{0\mid t})$ |
| $c_1$ | Armijo constant |
| $s$ | Line search backtracking factor |
| **Mathematical Notation** | |
| $\mathcal{N}(\mu, \Sigma)$ | Normal distribution with mean $\mu$ and covariance $\Sigma$ |
| $\|\cdot\|_2$ | L2 norm |
| $\nabla_{\boldsymbol{x}_t}$ | Gradient with respect to $\boldsymbol{x}_t$ |
| $I$ | Identity matrix |

## B    BROADER IMPACT

The widespread adoption of generative models across domains ranging from visual content creation (Rombach et al., 2022; Saharia et al., 2022; Blattmann et al., 2023) to audio synthesis (Kong et al., 2020; Liu et al., 2023) and 3D modeling (Zhang et al., 2023a; Vahdat et al., 2022; Wang et al., 2023) has fundamentally transformed creative and scientific workflows. The underlying diffusion framework (Ho et al., 2020; Song et al., 2020b) that powers these applications increasingly requires reliable alignment mechanisms to ensure generated content meets diverse user objectives and safety requirements across varied deployment contexts.

**Methodological Transparency and Responsible Development** Our reward-agnostic quality predictor approach enhances reproducibility and interpretability compared to black-box guidance methods, potentially supporting more responsible deployment practices. The explicit uncertainty quantification and principled optimization framework provide clearer insights into the guidance process, enabling better monitoring and evaluation of alignment outcomes (Adadi & Berrada, 2018; Rudin, 2019). This transparency facilitates the development of appropriate safeguards and content filtering mechanisms.

**Bias Amplification Through Better Alignment** More effective reward-based guidance may inadvertently amplify biases present in reward functions, training data, or evaluation metrics. Our uncertainty-aware optimization approach, while technically superior, could make biased objectives easier to achieve by providing more reliable alignment mechanisms. When reward functions encode societal biases or when training data contains systematic prejudices, improved alignment efficiency may strengthen rather than mitigate these problematic patterns in generated content (Mehrabi et al., 2021; Zhou et al., 2024).

## C    ADDITIONAL IMPLEMENTATION DETAIL

**Quality Predictor Training** We empirically compared different neural network architectures for the quality predictor, including MLPs, ResNets (He et al., 2016), and attention-based models (Vaswani et al., 2017). When training on latent representations, the MLP architecture demonstrated superior generalization performance with minimal computational overhead. Our final predictor uses an MLP with hidden dimensions [128, 256, 512], ReLU activations, and dropout (rate 0.1), achieving a mean squared error (MSE) of 0.017 with only 8M parameters. We constructed a 10K latent-similarity paired dataset using prompts from DiffusionDB (Wang et al., 2022) for training, with single training sessions completing in approximately 4 minutes and 40 seconds on a single RTX 4090 GPU.

**Other Implementation Details** For quantitative comparison, we used unified experimental settings with guidance scale (CFG) = 7.5, total timesteps $T = 50$, and $\eta = 1$. All experiments were conducted using text prompts from the HPSv2 benchmark (Wu et al., 2023). For DAS, we set the tempering parameter to 0.008 and number of particles to 4, consistent with the original paper. For other baseline methods, we adapted the official PyTorch codebase of DAS (Kim et al., 2025) to incorporate with the diffusers library. For our method, we consistently used $\kappa = 1.5$ for the LCB conservativeness parameter, Armijo constant $c_1 = 0.001$, ESS threshold $\delta = 0.8$, quantile constraint proportion $\rho = 0.8$ and line search backtracking factor $s = 0.5$ across all experiments. We also set the number of particles to 4 for a fair comparison. The code will be released after acceptance.

## D    ADDITIONAL ANALYSIS

### D.1    CHOICE OF VISUAL ENCODER: DINOv2 VS. CLIP

We empirically evaluated different pre-trained vision encoders for computing denoising quality labels and found that DINOv2-base (Oquab et al., 2023) provides superior sensitivity to denoising progress compared to multimodal encoders like CLIP (Radford et al., 2021), including clip-vit-base-patch32 and clip-vit-large-patch14 (Fig. 7). This performance difference stems from fundamental architectural distinctions in their training objectives and feature representations. DINOv2 employs a self-supervised learning approach optimized for discriminative visual understanding, where the CLS token is specifically trained to capture hierarchical visual features that distinguish between different structural qualities within images. In contrast, CLIP's contrastive learning framework optimizes

for cross-modal alignment between vision and language, producing representations that prioritize semantic similarity over fine-grained visual structure discrimination.

Specifically, Fig. 7 shows that CLIP embeddings exhibit limited discrimination in early denoising stages: when computing cosine similarity between intermediate predictions $\hat{x}_{0|t}$ and final outputs $x_0$, CLIP consistently produces relatively high similarity scores (mean $\approx 0.73$) across timesteps 1000-600, even when $\hat{x}_{0|t}$ contains predominantly noise. This saturation effect occurs because CLIP's multimodal latent space focuses on semantic content alignment rather than structural coherence, making it less sensitive to the gradual emergence of visual structure during denoising.

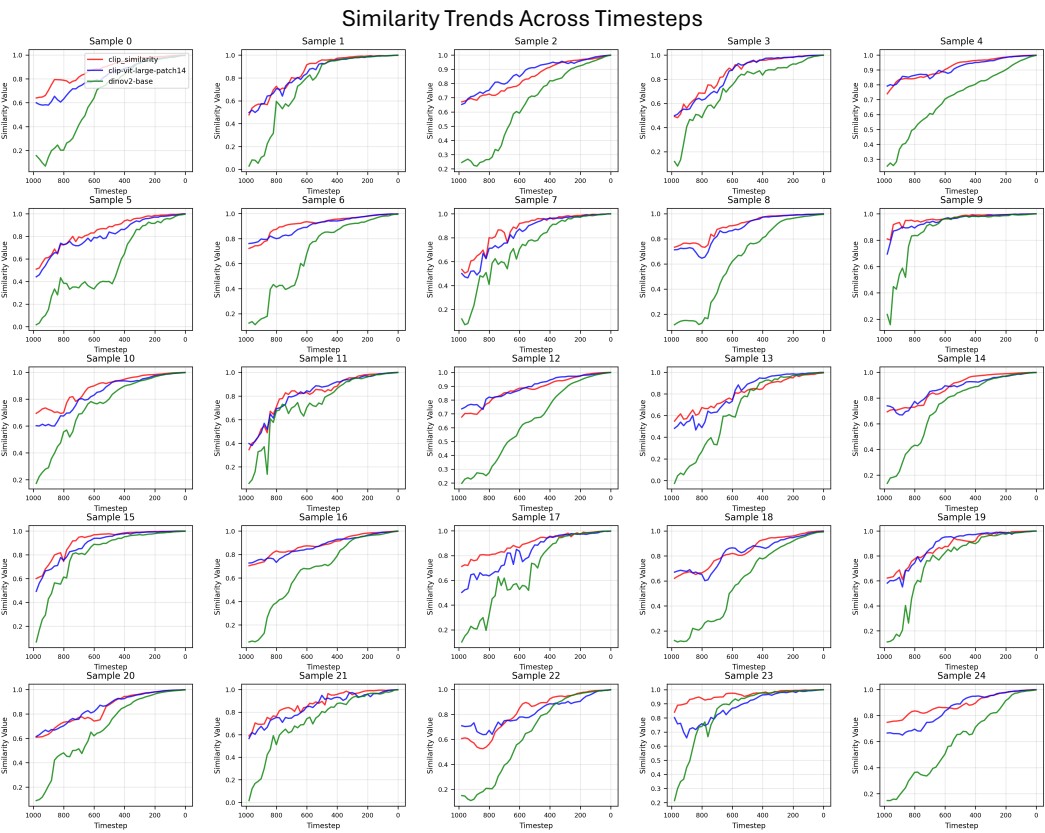

Figure 7: Comparison of visual encoder sensitivity to denoising progress across diffusion timesteps. Cosine similarity trajectories between intermediate predictions $\hat{x}_{0|t}$ and final outputs $x_0$ for 20 samples, computed using DINOv2-base, CLIP-ViT-Large-patch14, and CLIP-ViT-Base-patch32 encoders. DINOv2-base exhibits distinct trajectory variations across different samples, while CLIP variants show homogeneous patterns with limited sample-specific discrimination.

### D.2 IMPLICATIONS FOR LINE SEARCH ROBUSTNESS

The choice of quality predictor has important implications for the Armijo line search robustness. When the quality predictor's objective (structural coherence via DINOv2) differs significantly from the target reward function, the gradient directions $\nabla_{x_t} Q(x_t, t)$ and reward gradients $\hat{v} = \nabla_{x_t} \log R(\hat{x}_{0|t})$ may not be well-aligned. In such cases, the dot product $\nabla_{x_t} Q(x_t, t)^T \hat{v}$ in Definition 4.1 becomes smaller, making the Armijo sufficient decrease condition more difficult to satisfy. This naturally leads to more conservative step sizes, providing additional robustness when the quality predictor and target reward capture different aspects of sample quality. The reward-agnostic design of our quality predictor thus serves a dual purpose: enabling general applicability across diverse rewards while automatically promoting conservative guidance when predictor-reward alignment is uncertain.

### D.3 Validation of Quality Assessment

We provide additional validation through qualitative visualization and uncertainty analysis. Quantitative correlation analysis is presented in Sec. 5.3.

**Qualitative Validation.** Fig. 8 shows visual comparison of intermediate predictions $\hat{x}_{0|781}$ stratified by DINOv2 similarity scores, with both DINOv2 similarity and our predicted quality scores displayed on the top left corner. Samples with high DINOv2 similarity exhibit clear structural coherence and recognizable content, while low-similarity samples remain blurry and incoherent. Importantly, the predicted quality scores from our learned predictor closely track the DINOv2 similarity, demonstrating strong agreement between the two metrics. This visual correspondence mirrors the heterogeneous denoising progress observed in Fig. 1, confirming that our quality predictor successfully captures the actual structural quality variations that initially motivated our adaptive guidance approach.

**Uncertainty Validation for LCB-Based Tempering.** Our LCB-based tempering strategy (LCB $= \mu - \kappa\sigma$) relies on uncertainty estimates to identify high-risk regions requiring conservative exploration. We validate uncertainty quality through both relative and absolute calibration analyses.

**Relative Calibration.** For LCB-based ranking to be effective, uncertainty must correctly identify sample difficulty, i.e., higher predicted uncertainty should correlate with higher actual prediction error. We validate this property on 1,000 samples, confirming a positive correlation between predicted uncertainty and absolute error (Pearson $r = 0.41$, $p < 10^{-41}$). Stratified analysis reveals monotonic error increase across uncertainty quintiles, with samples in the highest uncertainty quintile (Q5, $\bar{\sigma} = 0.35$) exhibiting MAE=0.044, compared to MAE=0.024 for the lowest quintile (Q1, $\bar{\sigma} = 0.09$), a $1.8\times$ increase. This monotonic relationship validates that our uncertainty estimates provide meaningful relative rankings for identifying high-risk samples.

**Absolute Calibration.** To assess probabilistic calibration, we conduct coverage tests on the predicted confidence intervals. For our working hyperparameter $\kappa = 1.5$ (corresponding to an 87% confidence interval under Gaussian assumption), the actual coverage achieved is 64.9%. This $\sim$22% under-coverage is consistent with known trade-offs of Last Layer Laplace approximations, which prioritize computational efficiency over perfect probabilistic guarantees (Daxberger et al., 2021). Importantly, this under-coverage does not affect our method's effectiveness because: (1) LCB-based tempering relies on *relative rankings* rather than absolute confidence levels, and (2) the under-coverage is systematic and does not bias the relative ordering of samples. The consistent performance gains in Table 1 provide empirical evidence that these relative uncertainty signals effectively guide our adaptive tempering strategy, despite imperfect absolute calibration.

### D.4 Statistical Significance Analysis

To assess statistical robustness, we conducted 10 independent runs on 50 prompts (500 total samples) under Aesthetic guidance (Table 4). ImageReward shows significant improvement of +0.231 (+84%), with our method achieving less than half the standard deviation of DAS (0.04 vs. 0.10). The increased sample size confirms consistent gains on human-preference metrics (HPSv2: +0.8%, PickScore: +0.8%, CLIP: +1.8%) with substantially reduced variance, demonstrating statistical robustness of our approach.

Table 4: Statistical significance analysis with increased sample size (50 prompts $\times$ 10 runs). Standard deviations are computed across runs.

| Method | Aesthetic | CLIP | TCE | MPD | PickScore | HPSv2 | ImageReward |
|--------|-----------|------|-----|-----|-----------|-------|-------------|
| DAS | $6.1258_{\pm 0.06}$ | 0.2562 | $1.5131_{\pm 0.01}$ | $0.8499_{\pm 0.01}$ | 0.2182 | 0.2777 | $0.2752_{\pm 0.10}$ |
| SDAG | $6.0885_{\pm 0.04}$ | **0.2608** | $1.5057_{\pm 0.01}$ | **$0.8526_{\pm 0.01}$** | **0.2199** | **0.2798** | **$0.5066_{\pm 0.04}$** |

Table 5: Sensitivity analysis for $\delta$ (ESS threshold) and $\rho$ (quantile constraint). Performance remains stable across reasonable parameter ranges, with optimal results at $\delta = \rho = 0.8$.

| $\delta$ | Aesthetic | CLIP | TCE | MPD | PickScore | HPSv2 | ImageReward |
|---|---|---|---|---|---|---|---|
| 0.2 | $5.9555_{\pm 0.44}$ | $0.2575_{\pm 0.03}$ | 1.4912 | 0.8246 | $0.2200_{\pm 0.01}$ | $0.2796_{\pm 0.01}$ | $0.4371_{\pm 1.25}$ |
| 0.4 | $6.0587_{\pm 0.47}$ | $0.2578_{\pm 0.03}$ | 1.4805 | 0.8411 | $0.2200_{\pm 0.01}$ | $\mathbf{0.2808}_{\pm 0.01}$ | $0.5018_{\pm 1.11}$ |
| 0.6 | $6.0307_{\pm 0.48}$ | $0.2559_{\pm 0.03}$ | 1.4979 | 0.8549 | $0.2182_{\pm 0.01}$ | $0.2771_{\pm 0.02}$ | $0.3469_{\pm 1.14}$ |
| 0.8 | $\mathbf{6.1860}_{\pm 0.49}$ | $\mathbf{0.2645}_{\pm 0.03}$ | 1.5112 | 0.8558 | $\mathbf{0.2213}_{\pm 0.01}$ | $0.2800_{\pm 0.01}$ | $\mathbf{0.5989}_{\pm 1.20}$ |
| 1.0 | $6.1008_{\pm 0.58}$ | $0.2543_{\pm 0.03}$ | **1.5266** | **0.8562** | $0.2194_{\pm 0.01}$ | $0.2787_{\pm 0.01}$ | $0.3969_{\pm 1.16}$ |

(a) Varying $\delta$ (ESS threshold), fixing $\rho = 0.8$

| $\rho$ | Aesthetic | CLIP | TCE | MPD | PickScore | HPSv2 | ImageReward |
|---|---|---|---|---|---|---|---|
| 0.2 | $6.0907_{\pm 0.51}$ | $0.2601_{\pm 0.04}$ | 1.4981 | 0.8422 | $0.2199_{\pm 0.01}$ | $0.2799_{\pm 0.01}$ | $0.5521_{\pm 1.09}$ |
| 0.4 | $6.0107_{\pm 0.55}$ | $0.2585_{\pm 0.03}$ | 1.4998 | 0.8395 | $0.2209_{\pm 0.01}$ | $0.2798_{\pm 0.01}$ | $0.4636_{\pm 1.00}$ |
| 0.6 | $6.0166_{\pm 0.47}$ | $\mathbf{0.2693}_{\pm 0.03}$ | 1.5042 | 0.8517 | $0.2199_{\pm 0.01}$ | $0.2794_{\pm 0.01}$ | $0.5575_{\pm 1.10}$ |
| 0.8 | $\mathbf{6.1860}_{\pm 0.49}$ | $0.2645_{\pm 0.03}$ | **1.5112** | **0.8558** | $\mathbf{0.2213}_{\pm 0.01}$ | $\mathbf{0.2800}_{\pm 0.01}$ | $\mathbf{0.5989}_{\pm 1.00}$ |
| 1.0 | $6.0557_{\pm 0.47}$ | $0.2634_{\pm 0.03}$ | 1.4957 | 0.8399 | $0.2203_{\pm 0.01}$ | $0.2799_{\pm 0.01}$ | $0.4947_{\pm 1.15}$ |

(b) Varying $\rho$ (quantile constraint), fixing $\delta = 0.8$

## D.5 HYPERPARAMETER SENSITIVITY ANALYSIS

We conducted ablations to assess sensitivity to the ESS threshold $\delta$ and quantile constraint proportion $\rho$, varying each in $\{0.2, 0.4, 0.6, 0.8, 1.0\}$ while fixing the other at 0.8 (Table 5). Results show that performance is stable across reasonable parameter ranges. This stability stems from the roles of these parameters: $\delta$ prevents particle degeneracy (effective when $\delta \geq 0.6$), while $\rho$ ensures feasible global steps (effective when $\rho \geq 0.6$). Within these ranges, the dual-constraint mechanism (Definition 4.2) adaptively balances individual and population-level needs without requiring precise tuning. We chose $\delta = \rho = 0.8$ as a principled middle ground following SMC conventions.

## D.6 CROSS-ARCHITECTURE GENERALIZABILITY.

To validate our method's generalizability across different diffusion architectures, we evaluated SDAG on SD2.1-base and SDXL Podell et al. (2023) using Aesthetic guidance. Table 6 shows that SDAG maintains consistent advantages across both architectures. On SD2.1, SDAG achieves improvements on 5/7 metrics, with notable gains on ImageReward (+13.8%). On SDXL, SDAG demonstrates more substantial improvements with ImageReward gains of +18.5%, alongside consistent gains on human-preference metrics (HPSv2: +1.1%). Notably, SD2.1 and SD1.5 share compatible latent spaces, allowing us to apply the SD1.5-trained quality predictor directly to SD2.1 without retraining, demonstrating true plug-and-play generalizability across model versions.

Table 6: Cross-architecture evaluation on SD2.1-base and SDXL with Aesthetic guidance. SDAG maintains consistent improvements across different diffusion architectures, particularly on human-preference metrics. Best results in bold.

| | SD2.1-base | | | | | | |
|---|---|---|---|---|---|---|---|
| Method | Aesthetic | CLIP | TCE | MPD | Pick | HPS | IR |
| DAS | **6.15** | 0.255 | **1.511** | 0.828 | 0.220 | 0.281 | 0.49 |
| SDAG | 6.07 | **0.256** | 1.495 | **0.837** | **0.222** | **0.283** | **0.56** |

(a) SD2.1-base results

| | SDXL | | | | | | |
|---|---|---|---|---|---|---|---|
| Method | Aesthetic | CLIP | TCE | MPD | Pick | HPS | IR |
| DAS | 5.71 | 0.276 | **1.009** | 0.832 | 0.231 | 0.282 | 1.17 |
| SDAG | **5.73** | **0.280** | 0.982 | **0.837** | 0.231 | **0.285** | **1.39** |

(b) SDXL results

High Similarity          Low Similarity

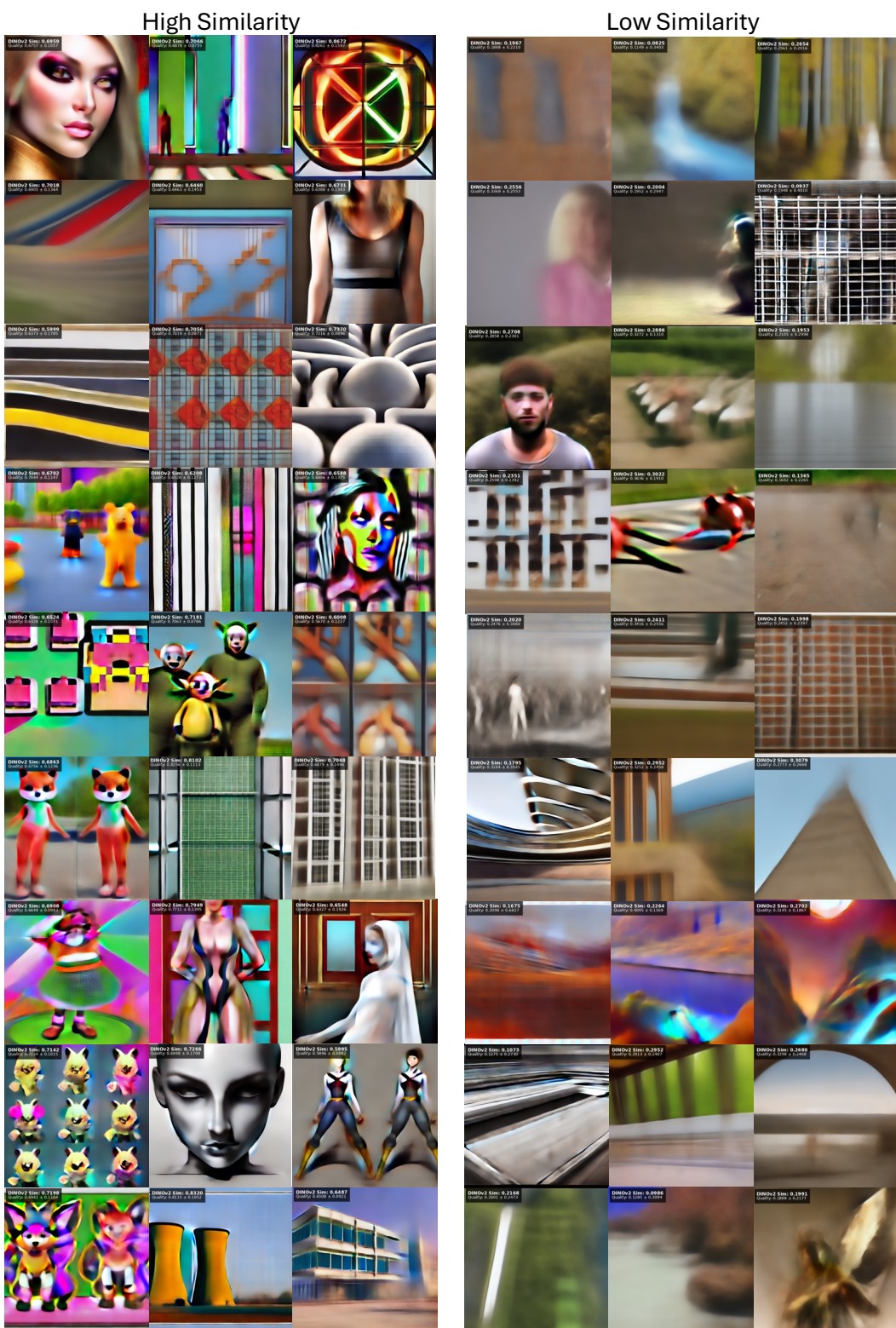

Figure 8: Visual comparison of intermediate predictions $\hat{x}_{0|781}$ stratified by DINOv2 similarity scores (DINOv2 similarity and our predicted score are shown on the top left corner). Left column shows samples with high DINOv2 similarity (clear structure and coherent content), while right column shows samples with low DINOv2 similarity (blurry, incomplete, or incoherent predictions). This demonstrates the quality predictor's ability to distinguish denoising progress at intermediate stages.

