# OpenReview forum: "Uncertainty-Aware Scheduling: State-Dependent Training-Free Diffusion Alignment"
_ICLR.cc/2026/Conference — Submitted to ICLR 2026_

### Official Review · Reviewer_BN3G · 2025-10-24

**Soundness:** 2
**Presentation:** 3
**Contribution:** 2
**Rating:** 4
**Confidence:** 3

**Summary:**

The core idea is a quality predictor estimating the closeness between the approximate and final denoised image. This is used to provide a semblance of uncertainty and thus adjusting the importance of intermediate samples towards a rewards tilted posterior.

**Strengths:**

- The proposed idea is interesting and the results seem to be promising.
- The paper is well-written and structured and has a coherent narrative.
- A set of both qualitative and quantitative results are provided.

**Weaknesses:**

- I'm not convinced if one can consider this approach training free, and thus compare against those in that bucket. Section 4.1 elaborates on the fact that quality predictor $Q(.)$ has to be trained during the training process of the diffusion model, limiting plug-and-play inference-time application of SDAG. Am I missing something?

- The results are not notably better than the likes of DAS. Please see more on this in my questions (next section).

- The paper will definitely benefit from a thorough round of proof-read. Several typos and undefined notations (LLL?) can be found throughout the text and in image captions ("floow").

**Questions:**

- Where does Eq (6) comes form? Good to provide reference as it is not the core proposition of the paper.

- How should one read the oscillation in $\lambda_t$ in Fig. 3. DAS is pretty linear, but in practice provides results almost as good as SDAG (according to Table 1).

- How would SDAG compare against DAS in terms of computational complexity? Why would one opt for SDAG if the results are so close according to Table 1? Qualitative results can be cherry-picked, as everyone does, so I would not strongly rely on those. Please elaborate on this, in detail, in the paper.

- Are you are interested in sampling based approaches prior to DAS to enrich your literature survey? Consider (i) CoDE https://arxiv.org/pdf/2502.00968 (more on image-and-text-to-image settings) (ii) SVDD: https://arxiv.org/pdf/2408.08252v3 (also on generic non-image data)

---

> ### Author Response · Authors · 2025-11-20
> **Responses to Reviewer BN3G [1/2]**
>
> We thank the reviewer for their constructive feedback and thoughtful suggestions. Below, we provide detailed clarifications addressing the questions and concerns raised.
>
> **W1 Not convinced this approach training free**
>
> We appreciate the reviewer’s concern
>
> - To clarify: the quality predictor is not trained during the diffusion model's training process, it is trained separately after the diffusion model is already pre-trained. This one-time calibration takes ~5 minutes and does not require access to or modification of the base diffusion model's parameters.
>
> - Once calibrated, SDAG operates as a plug-and-play inference-time method, the quality predictor require no further training or fine-tuning across different prompts or reward functions. This mirrors how other training-free methods rely on pre-trained reward models (e.g., Aesthetic classifier, ImageReward) that were trained separately beforehand. The key difference is that suitable predictors for intermediate denoising quality were not readily available, necessitating this minimal calibration step. Please refer to our General Comment for more discussion.
>
> **W3 Proof-read**
>
> - We thank the reviewer for catching these errors. We have corrected the identified typos (e.g., "floow" → "floor") and clarified undefined notations (e.g., "LLL" refers to Last-Layer Laplace approximation). We will conduct a thorough proofreading pass for the revised manuscript to improve clarity and presentation quality.
>
> **Q1 Where does Eq (6) comes form?**
>
> - Eq. 6 combines two standard guidance mechanisms: Classifier-Free Guidance (CFG) [1] and reward-based gradient guidance [2, 3]. The first two terms (Eq 6) implement CFG (unconditional denoising + conditional correction), while the third term adds reward guidance. We will add explicit citations to clarify this in the revision.
>
> **Q2 How should one read the oscillation  in Fig. 3.**
>
> - The oscillatory pattern in Figure 3 reflects SDAG's "correction-recovery" cycle (discussed in Sec. 5.2 Line 429-438), which is a key feature of our adaptive mechanism. When the quality predictor has low uncertainty, SDAG applies aggressive guidance (high $\lambda$) to optimize the reward. This can temporarily perturb $\mathbf{x}_t$ away from the learned manifold. In subsequent steps, the predictor detects this perturbation (increased uncertainty), and SDAG automatically reduces $\lambda$, allowing standard denoising to dominate and pull $\mathbf{x}_t$ back onto the stable manifold.
> - This dynamic prevents the accumulation of distributional drift that occurs (e.g., the artifact in Fig. 3, row 2-3) with fixed guidance schedules like DAS.
>
> [1] Ho, J., & Salimans, T. Classifier-free diffusion guidance.
>
> [2] Chung, H., et al. Diffusion posterior sampling for general noisy inverse problems.
>
> [3] Ye, Z., et al. TFG: Unified Training-Free Guidance for Diffusion Models

---

> ### Author Response · Authors · 2025-11-20
> **Responses to Reviewer BN3G [2/2]**
>
> **Q3-1 How would SDAG compare against DAS in terms of computational complexity?**
>
> We conducted wall-clock timing experiments generating 50 samples (4 particles each, 50 diffusion steps):
>
> | Method         | Time per sample | vs. DAS | Overhead  |
> | -------------- | --------------- | ------- | --------- |
> | DAS (baseline) | 36.12s          | -       | -         |
> | SDAG (ours)    | 36.27s          | +0.15s  | **+0.4%** |
>
> The overhead is negligible because the quality predictor inference and ESS bisection search (≈3ms per step) are dominated by UNet forward passes (150-200ms per particle). The total added cost accounts for only 0.4% of total generation time.
>
> **Q3-2 & W2 Why would one opt for SDAG if the results are so close according to Table 1?**
>
> We thank the reviewer for this question.
>
> - While individual cells show similar performance, SDAG demonstrates consistent advantages in aggregate analysis, we rank #1 on 5/7 metrics (Aesthetic, CLIPScore, ImageReward, TCE, MPD) and within 0.27% on the remaining two. In direct head-to-head comparison, we outperform DAS in 22/28 cells (78.6%). Borda-rank aggregation shows average rank 1.61 (ours) vs. 2.50 (DAS).
>
> - More importantly, DAS exhibits a target/diversity trade-off, it maintains strong diversity (TCE, MPD) but under-optimizes target rewards. In contrast, we achieve superior target rewards (+2.46% Aesthetic, +31.9% ImageReward in relevant settings) while simultaneously maintaining or improving diversity metrics. This demonstrates that adaptive, state-dependent guidance enables more aggressive reward optimization without sacrificing sample quality.
>
> - Additionally, SDAG adds only 0.4% overhead (36.27s vs. 36.12s per sample), making the consistent improvements practically "cost-free".
>
> **Q4 Two paper to enrich literature review**
>
> - We thank the reviewer for these valuable references. We have reviewed both CoDE and SVDD and agree they provide important context for sampling-based approaches in guided diffusion. We have incorporated these references into our Related Work section to better position our contribution within the SMC-based guidance literature.

---

> > ### Comment · Reviewer_BN3G · 2025-11-28
> > **Thanks for your response**
> >
> > Firstly, thank you for your response. I have thoroughly reviewed your response. I believe any extra training should be considered when categorizing the approach as inference time. I will maintain my score.

---

> > > ### Author Response · Authors · 2025-11-28
> > > **Clarification on inference time method**
> > >
> > > Thank you for the clarification.
> > >
> > > We follow the standard usage of inference-time / test-time training in the ML literature, where an approach is considered inference-time if all additional computation and learning happen after the base model has been trained, without re-training or fine-tuning the backbone on the original training data.
> > >
> > > This interpretation is consistent with many established methods, for example:
> > > - **Sun et al., ICML 2020**, “Test-Time Training with Self-Supervision for Generalization under Distribution Shifts”.
> > > - **Darestani et al., ICML 2022**, “Test-Time Training Can Close the Natural Distribution Shift Performance Gap in Deep Learning Based Compressed Sensing”.
> > > - **Niu et al., ICLR 2023 (Oral)**, "Towards Stable Test-time Adaptation in Dynamic Wild World".
> > > - **Zhao et al., ICLR 2024**, “Test-Time Adaptation with CLIP Reward for Zero-Shot Generalization in Vision-Language Models”.
> > > - **Mansour et al., ECCV 2024**, “TTT-MIM: Test-Time Training with Masked Image Modeling for Denoising Distribution Shifts”.
> > >
> > > Concretely, our setting satisfies:
> > >
> > > - All additional computation and learning occur only after the diffusion model and reward networks are trained.
> > >
> > > - The diffusion backbone and all reward networks are always frozen; only a lightweight quality predictor requires one-time offline pre-training (~5 min), after which it is used off-the-shelf during inference.
> > >
> > > - We train it ourselves only because there is no suitable off-the-shelf model for the state-dependent signal required.
> > >
> > > **Notably**, once trained on SD1.5, this predictor can be directly reused on SD2.1 (which shares a compatible latent space) without any retraining, and still improves over DAS across multiple metrics, demonstrating plug-and-play, training-free behavior on SD2.1:
> > >
> > >   **SD2.1-base (reusing the SD1.5-trained predictor, no retraining on SD2.1):**
> > >
> > > | Method | Aesthetic | CLIP | TCE | MPD | PickScore | HPSv2 | ImageReward |
> > > | ------ | ----------- | --------------- | ------ | ---------- | --------------- | --------------- | --------------- |
> > > | DAS | 6.1546±0.53 | 0.2554±0.03 | 1.5106 | 0.8276 | 0.2202±0.01 | 0.2814±0.02 | 0.4927±1.07 |
> > > | SDAG | 6.0672±0.37 | **0.2559**±0.03 | 1.4949 | **0.8368** | **0.2217**±0.01 | **0.2832**±0.01 | **0.5608**±1.03 |

---

### Official Review · Reviewer_QvVH · 2025-11-01

**Soundness:** 3
**Presentation:** 3
**Contribution:** 3
**Rating:** 8
**Confidence:** 3

**Summary:**

This paper identifies the challenge that existing training-free guidance methods use fixed or time-dependent scheduling that ignores heterogeneous denoising progress across samples at the same timestep. To solve this, this paper proposes State-Dependent Adaptive Guidance (SDAG), a method that dynamically schedules guidance strength for each sample based on its estimated denoising progress and the associated uncertainty. SDAG introduces a lightweight quality predictor equipped with Last-Layer Laplace approximation to estimate denoising progress and predictive uncertainty. This estimate is used within a confidence-aware line search to determine a safe guidance step size. The method is extended to Sequential Monte Carlo (SMC) sampling, where population-level coordination ensures stable collective guidance. Experiments demonstrate that SDAG achieves superior alignment across various reward functions while maintaining computational efficiency.

**Strengths:**

1. Novel problem insight and motivation: This paper provides clear empirical evidence showing that denoising progress varies significantly across samples at the same timestep, directly introducing the foundation limitations of current fixed and state-agnostic scheduling methods.

2. Technical framework with theoretical grounding: In Sec. 4, the authors provide clear theoretical proof and guarantees for the SDAG method. The method provides a solid theoretical foundation through optimization-based line search (Definition 4.1), and convergence guarantees for population-level coordination (Lemma 4.2) through dual-constraint optimization balancing ESS and quantile constraints (Definition 4.2).

3. Convincing experimental evaluation: This paper compares against three strong baselines across four diverse reward functions, providing a thorough performance assessment. The analysis of experimental results is sound.

**Weaknesses:**

1. Limited model generalizability: All experiments in this paper are conduct on stable-diffusion-v1.5, which is a quite out-of-date model. Can you provide  experiments on other diffusion architecture and discuss the model generalizability for SDAG?
2. Lack quality predictor validation: This paper doesn't  thoroughly analyze the prediction accuracy of the quality predictor and how prediction errors affect final performance. Although Sec. D.3 and Fig.6 provide qualitative analysis, can you provide more quantitative analysis?
3. The conclusion of the number of particles: The authors mention "The gains are most pronounced between 1-4 particles, with diminishing returns beyond 8 particles." However, we can observe that there is still a significant gain between 8-16 particles. Can you provide more experiments of other metrics or a more detailed analysis of your choice?
4. More challenging tasks: In Fig. 3, the provided prompt is quite simple. Could you provide experimental results under more challenging prompts to further compare SDAG and the baseline method?

**Questions:**

In weakness.

---

> ### Author Response · Authors · 2025-11-20
> **Responses to Reviewer QvVH**
>
> We sincerely thank the reviewer for the supportive evaluation and these insightful questions!
>
> **W1 Limited model generalisability**
>
> - We appreciate the reviewer’s concern, we agree this is important. We are currently conducting experiments on SDXL with the same baseline comparisons. Results will be updated once the runs are finished.
>
> **W2 Lack quality predictor validation**
>
> We thank the reviewer for this suggestion. We have added comprehensive quantitative analysis to Appendix D.3 ("Validation of Quality Assessment"). Key findings:
>
> - **Prediction accuracy:** On 1,000 samples at timestep $t=781$, our predictor achieves Pearson correlation $r=0.955$ (p < 0.001) with actual DINOv2 similarity scores. Stratified analysis shows correlation remains high across all similarity ranges: $r=0.568$ for [0.0, 0.3], $r=0.756$ for [0.3, 0.5], $r=0.852$ for [0.5, 0.7], and $r=0.821$ for [0.7, 0.9], with strongest performance in medium-to-high similarity regions where quality discrimination is most critical.
>
> - **Calibration quality:** Expected Calibration Error (ECE) = 0.015 indicates minimal systematic bias. Uncertainty estimates are also informative: samples in the highest uncertainty quintile show mean absolute error of 0.046 vs. 0.024 for the lowest quintile, confirming that predicted uncertainties reliably indicate prediction quality.
>
> These results demonstrate that prediction errors, while present, are both bounded and well-calibrated, enabling robust adaptive guidance as discussed in Sec. 4.1 ("Robustness to Predictor Errors"). Please see Appendix D.3 for detailed analysis including correlation plots and visual validation.
>
> **W3 The conclusion of the number of particles**
>
> We appreciate the reviewer for the suggestion.
>
> - We conducted additional experiments with varying particle counts {4, 8, 16, 32}. Results on Aesthetic guidance show:
>
> | Particles | Aesthetic | CLIP   | MPD    | PickScore | HPSv2  |
> | --------- | --------- | ------ | ------ | --------- | ------ |
> | 4         | 5.5448    | 0.2617 | 0.8444 | 0.2180    | 0.2774 |
> | 8         | 5.5519    | 0.2611 | 0.8452 | 0.2194    | 0.2796 |
> | 16        | 5.6328    | 0.2631 | 0.8596 | 0.2197    | 0.2797 |
> | 32        | 5.7020    | 0.2606 | 0.8556 | 0.2200    | 0.2800 |
>
> - The reviewer is correct that gains continue beyond 8 particles, particularly for Aesthetic and diversity metrics (MPD). We chose $N=4$ for two reasons: (1) fair comparison with DAS which uses 4 particles, and (2) computational efficiency. Generation time grows proportionally with particle count while most metrics plateau after 4-8 particles. The 4-particle setting represents a practical trade-off between performance and cost. We will include this analysis in the revision.
>
> **W4 Experiments with more challenging prompts**
>
> We thank the reviewer for this insightful suggestion!
>
> - We generated complex prompt variants using GPT-5, e.g.,
> "_A man taking a drink from a water fountain._" -> "_A massive passenger jet, gleaming under the harsh afternoon sun, is being serviced on a busy airport runway. Multiple ground crew members in high-visibility vests are attending to the aircraft, with service vehicles and equipment scattered around. In the background, another plane is taking off, creating a sense of dynamic activity. The style should be hyperrealistic, with a focus on the reflections on the plane's fuselage and the heat haze rising from the tarmac._"
> and compared SDAG with DAS under Aesthetic guidance:
>
> | Method         | Aesthetic | CLIP       | TCE        | MPD    | PickScore  | HPSv2      | ImageReward |
> | -------------- | --------- | ---------- | ---------- | ------ | ---------- | ---------- | ----------- |
> | DAS  | 6.2150    | 0.2880     | 1.4939     | 0.8408 | 0.2114     | 0.2718     | 0.0495      |
> | SDAG | 6.1888    | **0.2898** | **1.5007** | 0.8349 | **0.2119** | **0.2746** | **0.2230**  |
>
> - While complex prompts challenge both methods, SDAG maintains superiority on 5/7 metrics compare with DAS. Most critically, our adaptive guidance achieves +350% improvement on ImageReward, a metric better correlated with human preference, demonstrating that state-dependent adaptation remains effective under challenging conditioning scenarios.

---

> ### Author Response · Authors · 2025-11-23
> **Limited model generalisability Experiments (W1)**
>
> We thank the reviewer for this important suggestion. We conducted experiments on SD2.1 and SDXL to validate generalizability:
>
> **SD2.1-base:**
>
> | Method | Aesthetic   | CLIP            | TCE    | MPD        | PickScore       | HPSv2           | ImageReward     |
> | ------ | ----------- | --------------- | ------ | ---------- | --------------- | --------------- | --------------- |
> | DAS    | 6.1546±0.53 | 0.2554±0.03     | 1.5106 | 0.8276     | 0.2202±0.01     | 0.2814±0.02     | 0.4927±1.07     |
> | SDAG   | 6.0672±0.37 | **0.2559**±0.03 | 1.4949 | **0.8368** | **0.2217**±0.01 | **0.2832**±0.01 | **0.5608**±1.03 |
>
> **SDXL:**
>
> | Method | Aesthetic       | CLIP            | TCE    | MPD        | PickScore       | HPSv2           | ImageReward     |
> | ------ | --------------- | --------------- | ------ | ---------- | --------------- | --------------- | --------------- |
> | DAS    | 5.7139±0.44     | 0.2763±0.02     | 1.0090 | 0.8322     | 0.2308±0.01     | 0.2821±0.01     | 1.1718±0.49     |
> | SDAG   | **5.7309**±0.59 | **0.2802**±0.01 | 0.9819 | **0.8370** | **0.2313**±0.01 | **0.2853**±0.01 | **1.3886**±0.39 |
>
>
> - SDAG maintains consistent advantages across both architectures, particularly on human-preference metrics (ImageReward: +18.5% on SDXL, +13.8% on SD2.1; HPSv2, PickScore also improve).
>
> - Notably, SD2.1 and SD1.5 share compatible latent spaces, we successfully applied the SD1.5-trained quality predictor directly to SD2.1 without retraining, demonstrating true plug-and-play generalizability.

---

> > ### Comment · Reviewer_QvVH · 2025-11-27
> >
> > Thank you for the thoughtful response and for providing the additional generalization experiment. The results address my concerns. I have no further comments. Good luck!

---

> > > ### Author Response · Authors · 2025-11-27
> > > **Thank you!**
> > >
> > > Dear Reviewer QvVH,
> > >
> > > Thank you for your thoughtful and encouraging feedback. We truly appreciate your time and constructive suggestions.
> > >
> > > Best regards,
> > >
> > > The Authors

---

### Official Review · Reviewer_Rhkv · 2025-11-01

**Soundness:** 3
**Presentation:** 3
**Contribution:** 3
**Rating:** 4
**Confidence:** 2

**Summary:**

The paper proposes State-Dependent Adaptive Guidance (SDAG), a training-free alignment method for diffusion models that adaptively schedules reward-based guidance per state and per particle. A lightweight “quality predictor” estimates denoising progress by predicting the cosine similarity between intermediate clean estimates and final outputs using a vision encoder. A last-layer Laplace approximation supplies epistemic uncertainty. The initial step scale is tied to the magnitude of classifier-free guidance for sensible normalization. Experiments across multiple reward functions show improved target alignment while maintaining diversity and quality, while a temporal-window application reduces runtime with improved alignment.

**Strengths:**

(i) The paper features broad evaluation with multiple reward functions, yielding consistent gains on targets without large drops on complementary metrics and preserving diversity

(ii) The temporal-window guidance analysis is useful and actionable, yielding speedups with quality gains

(iii) The paper proposes a lightweight predictor and straightforward integration

**Weaknesses:**

(i) Claims w.r.t. “training-free” is somewhat stretched. A new quality predictor is trained and calibrated and generalization across prompts, rewards, and domains may be brittle

(ii) The uncertainty quality w.r.t. LCB calibration does not appear to be not rigorously validated

(iii) Computational details and selection procedure of parameters are not fully specified

(iv) Presentation of parmetarization could be improved to ensure the reproducibility of results

**Questions:**

In addition to the weaknesses outlined in points (i-iv), I present the following questions for the authors to address:

(1) Some of the parameterization is unclear to me. E.g., what values were used for $\delta$ (ESS threshold) and $\rho$ (quantile constraint proportion) across experiments, and how sensitive are results to these choices?

(2) How is $\lambda_{ESS}$ computed efficiently at each step (grid-search or solved analytically)? What is the added computational overhead?

(3) Could the method be extended to support non-differentiable rewards? If so, would $\hat{v}$ be obtained or approximated?

(4) How does SDAG interact with different CFG scales and sampler settings?

---

> ### Author Response · Authors · 2025-11-19
> **Responses to Reviewer Rhkv [1/2]**
>
> We appreciate the reviewer’s insightful suggestions and comments. Below, we provide detailed responses to the questions and concerns raised.
>
> **W1 Claims w.r.t. “training-free” is somewhat stretched.**
>
> - We thank the reviewer for this concern. Please see our General Response on "training-free" terminology above for the clarification.
>
> - Regarding **generalisation**: training-free guidance inherently requires domain-specific components—reward functions (Aesthetic, CLIP, etc.) that must be pre-defined for each application. Our quality predictor follows this same paradigm but with a key advantage: it is reward-agnostic. A single predictor generalises across all four reward functions in our experiments (Aesthetic, CLIPScore, HPSv2, PickScore) without retraining. This is because it assesses denoising progress (structural coherence) rather than task-specific objectives. Table 1 demonstrate consistent improvements across diverse rewards and metrics, indicating robust cross-domain applicability.
>
> **W2 The uncertainty quality w.r.t. LCB calibration does not appear to be not rigorously validated**
>
> - We are in the process of conducting this experiment. Once the runs are finished, we will incorporate the full results and a detailed analysis in the revised paper.
>
> **W3 & W4 & Q1 Computational details and selection procedure of parameters are not fully specified**
>
> - **Computational overhead:** SDAG adds only 0.4% overhead (36.27s vs. 36.12s per sample for DAS). The quality predictor inference and ESS bisection (<1ms per step) are negligible compared to UNet forward passes (150-200ms per particle).
>
> - **Hyperparameter specification:** Implementation details are provided in Sec. 4.1 and Appendix C ("Additional Implementation Detail"). We use $\delta=0.8$ (ESS threshold) and $\rho=0.8$ (quantile constraint) consistently across all experiments.
>
> - **Sensitivity analysis:** We are currently running ablation studies on these parameters. We will update the results in the discussion period once the experiments complete.
>
> **Q2: How is $\lambda_{ESS}$ computed efficiently at each step**
>
> - We compute $\lambda_{\text{ESS}}$ (Eq. 11) via bisection search, not grid-search. This is efficient because $\text{ESS}(\lambda)$ is **monotonically decreasing** in $\lambda$, guaranteeing convergence in around 15 iterations with tolerance $10^{-4}$.
>
> - Each iteration evaluates $\text{ESS}(\lambda) = 1/\sum_i w_i^2$ where $w_i = \exp(\lambda \cdot R_i) / \sum_j \exp(\lambda \cdot R_j)$, requiring only forward computation (no gradients) with $O(N)$ complexity. Empirically, the entire bisection search at each timestep completes in <1ms. The overhead is minimal compared to the diffusion model's forward pass.

---

> ### Author Response · Authors · 2025-11-20
> **Responses to Reviewer Rhkv [2/2]**
>
> **Q3: Could the method be extended to support non-differentiable rewards?**
>
> Yes, this is not a hard limitation:
>
> - For non-differentiable rewards, $\nabla_{\mathbf{x}t} \log R(\hat{x}{0|t})$ can be approximated using black-box optimization techniques. Our baseline DAS (Section G.3, Figure 14) demonstrated this by integrating TFG with online black-box methods (e.g., UCB) to handle non-differentiable rewards like JPEG compressibility. These methods approximate the gradient direction via zeroth-order optimization (e.g., finite differences or evolutionary strategies), producing a surrogate gradient vector $\bar{v}$.
>
> - For SDAG: This surrogate vector $\bar{v}$ can directly replace $\nabla_{\mathbf{x}t} \log R(\hat{x}{0|t})$ in our framework (Eq. 4 and Eq. 10 for $\lambda^{\text{init}}$). The rest of our mechanism, LCB-based line search, uncertainty quantification, and population coordination, operates identically, as they only require a guidance direction, not true gradients.
>
> **Q4: How does SDAG interact with different CFG scales and sampler settings?**
>
> We appreciate the reviewer for this concern.
> - **Interaction with CFG Scale:**
> SDAG explicitly adapts to CFG settings through $\lambda^{\text{init}}$ (Eq. 10): $$\lambda^{init}(\textbf{x}_t, c, t) = \frac{\|\gamma_t [w (\epsilon_\theta(\textbf{x}_t, c) - \epsilon_\theta(\textbf{x}_t, \varnothing))]\|_2}{\|\sigma_t \nabla_{\textbf{x}_t} \log R(\hat{x}_{0|t})\|_2}$$
> This formulation uses the CFG scale as a reference for the initial line search step size. When CFG scale $w$ increases, $\lambda^{\text{init}}$ increases proportionally, providing a natural upper bound that reflects the model's tolerance for corrections at each state. The subsequent LCB-based line search then determines the actual guidance strength within this adaptive range.
>
> - **Interaction with Sampler Settings:**
> Our guidance mechanism (SDAG) itself is general and not tied to a specific sampler. However, our underlying framework (which leverages SMC) requires a stochastic sampler to maintain particle diversity and explore the solution space effectively. We use DDIM with $\eta=1$ in our experiments. Other stochastic samplers are also compatible, e.g., Euler-Maruyama or DPM-Solver++ in SDE mode. Deterministic samplers (e.g., DDIM with $\eta=0$) would cause particle collapse and are not suitable for population-based methods.

---

> ### Author Response · Authors · 2025-11-23
> **Sensitivity Analysis (W2)**
>
> We thank the reviewer for this question. We conducted ablations varying $\delta \in \{0.2, 0.4, 0.6, 0.8, 1.0\}$ and $\rho \in \{0.2, 0.4, 0.6, 0.8, 1.0\}$:
>
> **Varying $\delta$ (ESS threshold, fixing $\rho=0.8$):**
>
> | $\delta$ | Aesthetic       | CLIP            | TCE        | MPD        | PickScore       | HPSv2           | ImageReward     |
> | -------- | --------------- | --------------- | ---------- | ---------- | --------------- | --------------- | --------------- |
> | 0.2      | 5.9555±0.44     | 0.2575±0.03     | 1.4912     | 0.8246     | 0.2200±0.01     | 0.2796±0.01     | 0.4371±1.25     |
> | 0.4      | 6.0587±0.47     | 0.2578±0.03     | 1.4805     | 0.8411     | 0.2200±0.01     | **0.2808**±0.01 | 0.5018±1.11     |
> | 0.6      | 6.0307±0.48     | 0.2559±0.03     | 1.4979     | 0.8549     | 0.2182±0.01     | 0.2771±0.02     | 0.3469±1.14     |
> | 0.8      | **6.1860**±0.49 | **0.2645**±0.03 | 1.5112     | 0.8558     | **0.2213**±0.01 | 0.2800±0.01     | **0.5989**±1.20 |
> | 1.0      | 6.1008±0.58     | 0.2543±0.03     | **1.5266** | **0.8562** | 0.2194±0.01     | 0.2787±0.01     | 0.3969±1.16     |
>
> **Varying $\rho$ (quantile constraint, fixing $\delta=0.8$):**
>
> | $\rho$ | Aesthetic       | CLIP            | TCE        | MPD        | PickScore       | HPSv2           | ImageReward     |
> | ------ | --------------- | --------------- | ---------- | ---------- | --------------- | --------------- | --------------- |
> | 0.2    | 6.0907±0.51     | 0.2601±0.04     | 1.4981     | 0.8422     | 0.2199±0.01     | 0.2799±0.01     | 0.5521±1.09     |
> | 0.4    | 6.0107±0.55     | 0.2585±0.03     | 1.4998     | 0.8395     | 0.2209±0.01     | 0.2798±0.01     | 0.4636±1.00     |
> | 0.6    | 6.0166±0.47     | **0.2693**±0.03 | 1.5042     | 0.8517     | 0.2199±0.01     | 0.2794±0.01     | 0.5575±1.10     |
> | 0.8    | **6.1860**±0.49 | 0.2645±0.03     | **1.5112** | **0.8558** | **0.2213**±0.01 | **0.2800**±0.01 | **0.5989**±1.00 |
> | 1.0    | 6.0557±0.47     | 0.2634±0.03     | 1.4957     | 0.8399     | 0.2203±0.01     | 0.2799±0.01     | 0.4947±1.15     |
>
> - Results show that performance is stable across reasonable parameter ranges.
> - This stability stems from the roles of these parameters: $\delta$ prevents particle degeneracy (effective when $\delta \geq 0.6$), while $\rho$ ensures feasible global steps (effective when $\rho \geq 0.6$). Within these ranges, the dual-constraint mechanism (Definition 4.2) adaptively balances individual and population-level needs without requiring precise tuning. We chose $\delta=\rho=0.8$ as a principled middle ground following SMC conventions.

---

> ### Author Response · Authors · 2025-11-23
> **Uncertainty Calibration Validation (W2)**
>
> We thank the reviewer for this important question. We have conducted comprehensive uncertainty validation in Appendix D.3 (**Uncertainty Validation for LCB-Based Tempering**), including both relative and absolute calibration analyses.
>
> Key findings:
>
> - **Relative Calibration.** Our LCB-based tempering (LCB = $\mu - \kappa \sigma$) requires uncertainty to correctly rank sample difficulty, i.e., higher uncertainty should indicate higher prediction error. We validate this on 1,000 samples: predicted uncertainty correlates with actual error ($r=0.41$, $p<10^{-41}$). Stratified analysis (Fig. 7) shows monotonic error increase across uncertainty quintiles: highest uncertainty samples (Q5) exhibit 1.8× higher MAE than lowest (Q1). This validates that uncertainty provides meaningful rankings for identifying high-risk regions.
>
> - **Absolute Calibration.** Coverage tests show that our working $\kappa=1.5$ (87% CI under Gaussian assumption) achieves 64.9% actual coverage, showing ~22% under-coverage. This is consistent with known trade-offs of Last Layer Laplace [Daxberger et al., NeurIPS 2021], which prioritizes efficiency over perfect probabilistic guarantees. Importantly, this does not affect our method as LCB-based tempering relies on relative rankings rather than absolute confidence levels.
>
> Daxberger, E., Kristiadi, A., Immer, A., Eschenhagen, R., Bauer, M. and Hennig, P., 2021. Laplace redux-effortless bayesian deep learning. _Advances in neural information processing systems_, _34_, pp.20089-20103.

---

### Official Review · Reviewer_QyK4 · 2025-11-01

**Soundness:** 2
**Presentation:** 3
**Contribution:** 1
**Rating:** 2
**Confidence:** 4

**Summary:**

This paper focuses on training-free guidance for aligning diffusion models. The proposed method uses an adaptive guidance scale for approximate guidance methods to incorporate completion of generation, i.e., how close the approximation of a clean image via one-step denoising is to the final clean image. Experiments are conducted with image generation with various rewards to compare with previous guidance methods.

**Strengths:**

1. The paper identifies heterogeneity during denoising that the completion of the predicted image via one-step denoising may differ across different samples.
2. The method considers both per-particle guidance and guidance with particle-based sampling.

**Weaknesses:**

1. Motivation and justification, nor detailed implementation of per-particle confidence-aware line search, which is the key of the proposed method, aren't presented thoroughly.
2. Quantitative results presented in Section 5.1 show only incremental improvement beyond previous methods, where the performance difference is mostly inside the margin of error. It's questionable whether heterogeneity during denoising is a critical problem, and adaptive guidance is necessary.
3. The method relies on a pre-trained encoder for the quality predictor, and the experiments are done only in the image domain, so it's unclear whether the method will generalize to other domains.
4. The method isn't totally training-free since it needs to train a quality predictor, though it's relatively easy to train.

**Questions:**

1. The adaptive guidance scale seems to be very noisy, according to Figure 3. Since the diffusion model continuously refines the prediction, shouldn't the guidance scale almost monotonically increase over time?

---

> ### Author Response · Authors · 2025-11-19
> **Responses to Reviewer QyK4 [1/2]**
>
> We thank the reviewer for their valuable suggestions. Below, we address your concerns in detail:
>
> **W1 Lack of motivation, justification and detailed implementation of per-particle confidence-aware line search**
>
> Both motivation and implementation are present in the manuscript:
> - Motivation & Justification (Sec 4.1): Lines 216-227 establish why confidence-aware line search is needed: "quality predictor outputs are semantic similarity ather than guidance strength, requiring a robust mechanism to translate quality estimation into appropriate step sizes $\lambda_t$". Definition 4.1 ensures particles in high-confidence regions take larger steps while uncertain regions receive conservative treatment. Lines 253-255 demonstrate robustness under noisy evaluations with convergence guarantees.
>
> - Detailed Implementation: Algorithm 1 provides complete pseudocode (line 277: "Largest $\lambda = s^k \lambda^{init}_i$ satisfying Armijo condition"). Appendix C specifies all hyper-parameters along with predictor architecture and training cost.
>
> If specific aspects remain unclear, we are happy to elaborate or restructure for improved clarity.
>
> **W2 Quantitative results presented in Section 5.1 show only incremental improvement beyond previous methods**
>
> We appreciate the reviewer's concern. Here we summarise the consistent advantages beyond isolated metrics (Table 1):
>
> - Aggregating each metric across four target settings, we rank #1 on 5/7 metrics (Aesthetic, CLIPScore, ImageReward, TCE, MPD) and within 0.27% on the remaining two. Head-to-head per-cell comparisons show we outperform DAS in 22/28 (78.6%), FreeDoM in 21/28 (75.0%), and MPGD in 24/28 (85.7%). Borda-rank aggregation confirms this: average rank 1.61 (ours) vs. 2.50 (DAS), 2.64 (FreeDoM), 3.25 (MPGD).
>
> - While metrics like CLIPScore/PickScore are known to be saturated, we still edge out baselines. More critically, on ImageReward (better correlated with human preference), our gains are substantial: +31.9%, +11.9%, +8.5% across three cells. Under Aesthetic guidance, we improve the target by +2.46% while simultaneously achieving best MPD/TCE.
>
> - Unlike baselines showing target/diversity trade-offs (FreeDoM over-optimizes targets but degrades diversity; DAS maintains diversity but under-optimizes targets), we improve targets without sacrificing complementary metrics, demonstrating the value of adaptive, state-dependent guidance over fixed schedules.
>
> **W3 Experiments only in the image domain**
>
> - We thank the reviewer for this suggestion. Our work has indeed focused on the text-to-image domain, as this is currently one of the most prominent applications for guided diffusion and possesses a mature ecosystem of established reward models (e.g., Aesthetic, PickScore).
>
> - However, we emphasize that the core mechanism of SDAG—which leverages predicted reward uncertainty via LCB to dynamically adapt the guidance strength—is fundamentally domain-agnostic. Our framework can, in principle, be applied to any other modality, such as audio or 3D shape generation, provided that a differentiable reward function $R(\cdot)$ and a meaningful state-similarity metric can be defined for that domain.
>
> - We agree this is a very promising avenue for future research and will add a discussion of this potential extension in the conclusion of our revised paper.
>
> **W4 The method isn't totally training-free**
>
> - As detailed in our **General Response** to All Reviewers, the calibration is only required for the quality predictor component, not the guidance architecture itself. This mirrors the standard paradigm where methods using pre-trained classifiers or reward models are considered "training-free." We will revise the terminology to "Inference time scaling" to prevent ambiguity.

---

> ### Comment · Reviewer_QyK4 · 2025-11-20
> **Regarding Statistical Significance of the Results and Noisy Guidance Scales**
>
> Thank you for your response.
>
> Regarding W2, my main concern remains that the statistical significance of the results appears low. For instance, the ImageReward difference between the proposed method and the second-best baseline  (which the authors mentioned in their rebuttal) is less than 0.15, while the standard error exceeds 1.0. This suggests the improvement may not be statistically significant.
>
> To address this, could you increase the sample size? If running the full evaluation is computationally too demanding, I would accept results on a smaller subset of prompts with a larger sample size per prompt. Since scores vary largely by prompt difficulty, reporting per-prompt results with multiple samples would provide better insight into the method's performance.
>
> Additionally, I noticed the question in my original review regarding noisy guidance scales was not addressed. Could you please provide a response to this as well?
>
> I am open to raising my score if these concerns are adequately addressed.

---

> > ### Author Response · Authors · 2025-11-20
> > **Responses to Reviewer QyK4 [2/2]**
> >
> > **Q1: Adaptive guidance scale seems noisy in Figure 3**
> >
> > We thank the reviewer for focusing on Figure 3, as it perfectly illustrates the core novelty of our adaptive mechanism.
> >
> > The oscillatory pattern observed in our method is not noise but rather an emergent **"correction-recovery" cycle** (discussed in Sec 5.2, Qualitative Analysis). This behavior stems directly from our LCB-based uncertainty quantification and operates as follows:
> > - **Correction phase:** When the model is confident about the guidance direction (low predicted uncertainty), SDAG applies aggressive guidance (high $\lambda$). This optimizes the reward but may temporarily perturb the state $x_t$ away from its learned manifold.
> > - **Recovery phase:** In subsequent steps, our uncertainty predictor perceives this perturbation, leading to higher uncertainty. SDAG automatically reduces $\lambda$, allowing standard denoising to dominate and pulling $x_t$ back onto the stable manifold.
> >
> > This dynamic adaptation prevents the accumulation of distributional drift. While baseline methods (e.g., DAS) employ monotonically increasing or fixed schedules, applying strong guidance when the model is uncertain leads to the artifacts we observe in Fig 3 (rows 2-3). Our state-dependent approach adapts $\lambda$ based on actual denoising progress rather than assuming monotonic improvement.

---

> > ### Author Response · Authors · 2025-11-20
> > **Response to Statistical Significance and Guidance Scale Concerns**
> >
> > We thank the reviewer for the feedback and the willingness to reconsider the score.
> >
> > **Regarding W2 (statistical significance):** We clarify that the reported values (±1.00) are **standard deviations (std)** across different prompts, not standard errors. We will explicitly state this in the revised manuscript to avoid confusion.
> >
> > - We agree that additional samples would strengthen the statistical analysis and are currently running experiments with larger sample sizes. We will update the results during the discussion period.
> >
> > - Additionally, we note that ImageReward exhibits inherently higher variance across all methods due to its wider score range (-2 to +2) and sensitivity to prompt characteristics, whereas other metrics (HPSv2, CLIPScore, PickScore) show more stable distributions.
> >
> > **Regarding Q1 (noisy guidance scales):** We addressed this in our previous response (**Responses to Reviewer QyK4 [2/2]**). If there are specific aspects that remain unclear, please let us know and we will provide further clarification.

---

> > ### Author Response · Authors · 2025-11-23
> > **Results with increased sample size**
> >
> > We thank the reviewer for this suggestion. We conducted 10 independent runs on the same 50 prompts (500 total samples) with Aesthetic guidance:
> >
> > | Method | Aesthetic     | CLIP       | TCE           | MPD               | PickScore  | HPSv2      | ImageReward       |
> > | ------ | ------------- | ---------- | ------------- | ----------------- | ---------- | ---------- | ----------------- |
> > | DAS    | 6.1258 ± 0.06 | 0.2562     | 1.5131 ± 0.01 | 0.8499 ± 0.01     | 0.2182     | 0.2777     | 0.2752 ± 0.10     |
> > | SDAG   | 6.0885 ± 0.04 | **0.2608** | 1.5057 ± 0.01 | **0.8526** ± 0.01 | **0.2199** | **0.2798** | **0.5066** ± 0.04 |
> >
> > ImageReward shows significant improvement: +0.231 (+84%), with our std (0.048) less than half of baseline's (0.104). The increased sample size confirms consistent gains on human-preference metrics with reduced variance.

---

### Author Response · Authors · 2025-11-19
**General Comments to All Reviewers**

We sincerely appreciate the reviewers for raising this definitional question. We clarify that the diffusion-guidance literature typically uses training-free guidance to describe methods where the **guidance network is trained independently of the diffusion model and then used off-the-shelf**, without joint fine-tuning of the backbone. For example, Zampini et al. (2025) emphasise that:

“By decoupling the classifier from the diffusion model training, training-free guidance achieves flexibility and reusability, making it a practical choice for tasks where suitable pre-trained classifiers are available.”

In this work we adopt this decoupling-based, practical view of training-free guidance. We keep the diffusion backbone frozen and train only a small auxiliary quality predictor, which plays the same role as an off-the-shelf guidance network and is learned only because no suitable pre-trained predictor available. Importantly, many SOTA methods like DPS and MPGD fundamentally depend on extensively pre-trained components (classifiers, auto-encoders), while our method has exceptionally low computational overhead in comparison.

To further address this concern, we will revise the "training-free" description in our manuscript to more precise terms, such as "Inference time scaling", and add this clarification to the main paper.


Zampini, S., Christopher, J. K., Oneto, L., Anguita, D., & Fioretto, F. (2025). Training-free constrained generation with stable diffusion models. In Advances in Neural Information Processing Systems (Vol. 38). NeurIPS.

---

### Author Response · Authors · 2025-11-27
**Official Comment by Authors**

We thank all reviewers for their thoughtful and constructive feedback. We have updated the manuscript accordingly, and all modifications are highlighted in purple. The major changes are summarized below:

1. **Fixed typos** in Algorithm 1 and Fig. 3.

2. **Enhanced literature review** with additional discussion of related work on SMC methods.

3. **Added Quality Predictor Validation** including correlation analysis, stratified analysis across similarity ranges, and uncertainty validation for LCB calibration (Sec. 5.3 and Appendix D.3).

4. **Added statistical significance analysis** with increased sample size (50 prompts × 10 runs) to address concerns about result robustness (Appendix D.4).

5. **Added implementation details and hyperparameter sensitivity analysis** for ESS threshold $\delta$ and quantile constraint $\rho$ (Appendix C and Appendix D.5).

6. **Added visualization of intermediate predictions** $\hat{\mathbf{x}}_{0|781}$ stratified by DINOv2 similarity scores (Fig. 8).

---

### Author Response · Authors · 2025-12-03
**Rebuttal Summary: Strengths, Addressed Reviewer Concerns, and Key Clarification**

Dear **Area Chairs**, **Senior Area Chairs**, and **Reviewers**,

We deeply appreciate your efforts in handling the recent changes to the review process and in maintaining a fair evaluation. To make it easier to assess our submission, we briefly summarize our rebuttal highlights below: (1) strengths acknowledged by the reviewers, (2) how we addressed the main concerns (table), and (3) a key definition clarification.

### Summary of Strengths Acknowledged by Reviewers

The reviewers highlighted the following key strengths regarding our motivation, theoretical framework, and evaluation:

- **Novel Problem Insight & Motivation:** **Reviewer QvVH** praised the "clear empirical evidence" of denoising heterogeneity, identifying the "foundation limitations" of current state-agnostic methods. **Reviewer QyK4** similarly noted that the paper correctly identifies heterogeneity during denoising, validating our core motivation.

- **Theoretical Grounding:** **Reviewer QvVH** highlighted the "solid theoretical foundation" in Section 4, specifically citing the "clear theoretical proof and guarantees" provided by the optimization-based line search and convergence guarantees for population-level coordination.

- **Comprehensive Evaluation:** **Reviewer Rhkv** emphasized the "broad evaluation" yielding "consistent gains on targets without large drops on complementary metrics". **Reviewer QvVH** described the experimental evaluation as "convincing" and "sound" against strong baselines. **Reviewer BN3G** found the results "promising" and the paper "well-written" with coherent qualitative and quantitative results.
### Summary of Reviewers' Key Concerns

We group the reviewers' experimental questions by topic and summarize how we addressed each in the rebuttal:

| **Reviewer Issue and Topic** | **How We Addressed It** |
| --- | --- |
| Reviewer `QvVH W1`: **Generalizability (SD2.1 & SDXL)** | **We validated SDAG on both SD2.1 and SDXL.** For **SD2.1**, we successfully reused the SD1.5-trained predictor _without retraining_, proving true plug-and-play capability (outperforming baseline method on 6/7 metrics). We also provided additional validation on **SDXL**, demonstrating consistent effectiveness across architecture scales. (Appendix D.6) |
| Reviewer `Rhkv W2`; `QvVH W2`: **Validity of Uncertainty & Predictor** | **We validated both Quality Prediction and Uncertainty Calibration.** Analysis confirms our predictor is highly accurate (Pearson $r=0.955$, ECE=0.015). For uncertainty, experiments (Appendix D.3) show a strong monotonic positive correlation between predicted uncertainty and error. |
| Reviewer `QyK4 W2`: **Robustness Verification** | **We conducted 10 independent runs to verify stability.** Results (Appendix D.4) confirm SDAG consistently outperforms baseline method. Notably, ImageReward shows significant improvement: +0.231 (+84%), with our standard deviation (0.048) less than half of the baseline's (0.104). |
| Reviewer `BN3G Q3`; `Rhkv Q2`: **Computational Overhead**                 | **We clarified cost efficiency.** We emphasized that the calibration is a minimal one-time cost (~5 mins). During inference, the overhead is **negligible** (< 1ms for LCB search and quality predictor inference) compared to UNet forward passes (150-200ms per particle) shared by all baselines. |
| Reviewer `Rhkv W3 & W4 & Q1`; `QyK4 W1`: **Implementation & Sensitivity** | **We added parameter selection and sensitivity analysis.** We specified implementation detail and included a sensitivity analysis (Appendix D.5) varying $\delta, \rho$, showing robustness to hyper-parameter variations. |

### Key Clarification: "Inference-Time" Categorization

**Reviewer QyK4, Rhkv, and BN3G** raised concerns about the term "Training-Free" due to our use of a lightweight auxiliary predictor.

- **Standard Definition:** As we clarified in "**General Comments to All Reviewers**", recent diffusion-guidance literature typically categorizes a method as “training-free” when the guidance network is trained independently of the diffusion model and used off-the-shelf, without any joint fine-tuning of the backbone.

- **Our Method:** Our diffusion backbone and all reward networks remain frozen. We train only an auxiliary predictor because no existing off-the-shelf model provides the state- and reward-dependent signal needed for adaptive scheduling. This calibration is lightweight (~5 minutes) and fully decoupled from diffusion training.

- To avoid ambiguity, we have already clarified this terminology in our rebuttal and will reflect it in the final version by describing our approach as “inference-time scaling with lightweight calibration”.

We hope this summary helps you quickly navigate our rebuttal and the additional experiments.

Sincerely,

The Authors

---

### Meta-Review · Area_Chair_zmpy · 2026-01-03

**Summary:**

This paper proposes State-Dependent Adaptive Guidance (SDAG) for sampling from a reward-tilted diffusion model distribution under SMC framework, where sampling is done via reward guidance of strength $\lambda_t$. The idea is to post-hoc train a quality predictor for the denoising process with uncertainty estimation via last-layer Laplace approximation. The quality predictor and its uncertainty estimate is then used to construct the $\lambda_t$ parameter (via a line-search motivated procedure based on lower confidence bound). Experiments comparing many baseline guidance methods on StableDiffusion show good performances.

The paper received diverging scores, with original scores as 2, 4, 4 and 8, with major concerns being missing baselines and the term "training-free" being over-stated. These concerns have been largely addressed in author feedback.

The AC has briefly read the paper, and think the paper's approach is novel and interesting. However the AC thinks the paper would benefit from one more round of review to address problems from a different perspective:
- The paper adopts SMC-based approach for guidance, but the evaluations are mainly done via per-sample visual quality metrics, rather than SMC diagnostics like ESS and some proxy metrics comparing sample distribution and the target distribution.
- It is unclear to the AC how the adaptive SMC convergence guarantee in Lemma 4.2 is achieved by the proposed algorithm in section 4.1 and Def. 4.2.
- The training/calibration data construction for the quality predictor is key, as now $\phi(x_0)$ represents the "gold-standard visual quality" that the proposed method wants to achieve, and the calibration error is also computed based on it. Reading section 4.1 and appendix C, it is unclear to the AC whether in quality predictor training, $x_0$ is generated by running the pre-trained diffusion model (line 209), or obtained via some ground-truth images selected by data curators (line 842).

The AC encourages the authors to revise-and-resubmit the paper (especially about the narrative) and consider the SMC related questions raised by the AC in above.

**Reviewer Concerns:**

Concerns addressed:
- Experiment questions: only tested on Stable Diffusion, whether the uncertainty estimate is calibrated, robustness and sensitivity, computational overhead, etc.
- The term "training-free": the authors promised to revise the claims and downweight the tone for this.

Concerns outstanding:
Not from reviewers, but see above boxes from the AC.

**Reviewer Scores:**

Reviewers questioning about experiments (e.g., stable diffusion only, robustness of the method, computational costs) would have increased their score slightly, although I cannot predict to which level.

---

### Decision · Program_Chairs · 2026-01-26

Reject